# Geochemical exploration of rare earth element resources in highland karstic bauxite deposits in the Sierra de Bahoruco, Pedernales Province, Southwestern Dominican Republic

**Mark Chappell**[1]*, **Harold Rojas**[2], **Charles Andros**[1], **Autumn Acree**[1], **Yoko Masue-Slowey**[1], **Christine Young**[1], **Paige Fowler**[1], **Elizabeth Lotufo**[1], **Wesley Rowland**[1], **Michelle Wynter**[1], **Marcelo Salles**[3], **Leopoldo Gonzalez**[2]

1 Environmental Laboratory, U.S. Army Engineer Research & Development Center, Vicksburg, Mississippi, United States of America, 2 Dirección General de Minería, República Dominicana, 3 Jacksonville District, U.S. Army Corps of Engineers, Jacksonville, Florida, United States of America

* mark.a.chappell@usace.army.mil

## Abstract

This study investigates the geochemical characteristics of rare earth elements (REEs) in highland karstic bauxite deposits located in the Sierra de Bahoruco, Pedernales Province, Dominican Republic. These deposits, formed through intense weathering of volcanic material, represent a potentially valuable REE resource for the nation. Surface and subsurface soil samples were analyzed using portable X-ray fluorescence (pXRF) and a NixPro 2 color sensor validated with inductively coupled plasma optical emission spectrometry (ICP-OES). We employed compositional data analysis (CoDA) and machine learning models to estimate total REE concentrations, demonstrating that pXRF and the color sensor, when properly calibrated, are effective tools for remote geochemical exploration. The results reveal that REE concentrations increase with depth and elevation, with light REEs (LREEs) dominating the profiles. The correlation of REE concentrations with morphological soil development suggests that higher-altitude areas are enriched in REEs due to progressive weathering processes. The study also shows a strong relationship between REE concentrations and environmental factors such as latitude and elevation. While pXRF provided reliable estimates of total REE concentrations, to our surprise, the NixPro2 color sensor proved similarly accurate. The research emphasizes the practical value of the x-ray and color sensors for remote exploration, provided that a well-explored, robust calibration is performed to account for site-specific variability. These findings contribute to understanding the geochemical distribution of REEs in karstic bauxite deposits and highlight the potential for further exploration in remote, high-altitude regions. Future research should explore using these and other portable sensors, singly or combined, to predict REE speciation, for expediting information related to the environmentally sustainable extractability and potential economic feasibility of resources in expeditionary locations.

**Data Availability Statement:** the data and coding notebooks are found on the following github site: https://github.com/candros/DR_REE_pXRF_Modeling.

**Funding:** The author(s) received no specific funding for this work.

**Competing interests:** The authors have declared that no competing interests exist.

# Introduction

REE represent 15 lanthanides (atomic numbers 57–71) and the elements scandium (Sc), yttrium (Y), and lanthanum (La) corresponding to atomic numbers 21, 39, and 57, respectively. REE are vial for modern technologies and play crucial roles in industrial, medical, and new "green-energy" applications. REE are widely used in radar and sonar applications, communications and heads-up displays (HUDs), high-capacity electric motors, and jet engines [1, and references therein], among other areas. Mechanistically, the large ionic size, unusual electromagnetic properties, strong physically adsorptive specificity and catalytic activity make REE highly valuable in technological development [1, and references therein, 2, and references therein, 3–7], making them highly sought globally.

Traditionally, economically viable REE concentrations are found in carbonatites, alkaline igneous settings, ion-adsorption clays, and monazite-xenotime-bearing placer deposits [8, 9]. However, this study explores REE concentrations and distribution in a less common environment–highly weathered karstic bauxite deposits within the Sierra de Bahoruco Mountains on the Bahoruco Peninsula in the Pedernales Province of the Southwestern Dominican Republic (DR).

The Sierra de Bahoruco Mountains were formed by the oblique convergence of the North American and Caribbean plates in Southern Hispaniola [10]. The Bahoruco Peninsula block is bounded by the Plantain Garden-Enriquillo fault zone to the north and the Eastern Beauta Ridge Fault to the east [11]. Uplift of the Sierra de Bahoruco Mountains likely began during the Middle Miocene and continues today [12].

The region's stratigraphy comprises the Campanian to lower Eocene Dumisseau Formation, overlain by Eocene to Quaternary carbonates [12]. The Dumisseau Fm. includes basaltic and pyroclastic flows and some sedimentary deposits, with a total thickness of 1.5 km. It represents the crystalline basement of the area, with a mantle plume identified as the source of Dumisseau basalt flows, which have been classified as low-Ti tholeiites, high-Ti transitional basalts and LREE-enriched alkaline basalts [13].

The region's carbonate sequences record the environmental transition that occurred before and after the uplift of the Sierra de Bahoruco. These sequences begin with deep, outer slope carbonates of the Aceitillar and Neiba Formations from the Eocene, followed by a regional unconformity separating them from the shallow inner platform and reef bound deposits of the Pedernales and Aguas Negras units (Oligocene–Pliocene). The final Quaternary carbonate sequences originated from lagoon and beach depositional environments, with repeated high sea-level stands (associated with sea-level oscillations) during the Pleistocene and Holocene forming karst landforms [14].

This study focuses on the baxuites in the Sierra de Bahoruco Mountains, which have been largely unexplored until recently. The term bauxite was first used by Berthier in 1821 to describe alumina-rich sediments in the Les Baux region of France [15]. Bauxite ore is primarily composed of Al (oxyhydr)oxides [15] along with variable amounts of Fe and Ti (oxyhydr)oxides. Bauxite deposits are classified into two main categories: those found on aluminosilicate bedrocks, and those on carbonate bedrocks, known as karst-type bauxite deposits [16, 17]. Recent studies have focused on the distribution and fractionation of REE in bauxite deposits worldwide, driven by the increasing demand for critical minerals. These studies suggest that REE distribution in karst bauxite deposits is influenced by several factors, including the chemical composition of the protolith and bedrocks [18], physicochemical conditions in the weathering environment (e.g., Eh–pH) [19–21], water table fluctuations [22], organic and inorganic ligands in the soil [21, 23], bacterial activity, mineral species in bedrocks [24, 25], adsorption and scavenging processes [26, 27], climatic conditions [28, 29], and chemical properties of

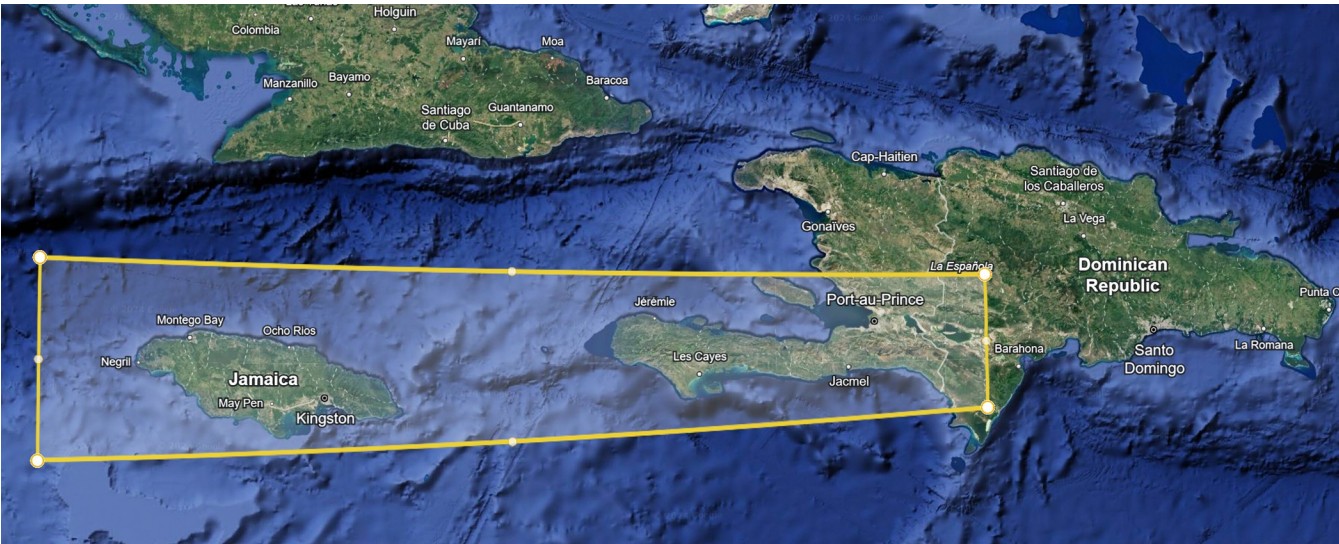

**Fig 1. Theoretical origin of the bauxite deposits is associated with ash deposition from ancient volcanic activity in Central America.** The yellow polygon demarcates the proposed zone of ash deposition, where the bauxite deposits exhibit geochemical similarities (adapted from Goldich, 1947).

elements during weathering [30]. Interestingly, evidence shows that karstic bauxites throughout the world generally contain higher REE concentrations than the more convenational lateritic bauxites [31].

The bauxites of the Sierra de Bahoruco Mountains are believed to have originated from either ash deposits from explosive Miocene volcanic eruptions in Central America [32, 33] or from post-Pliocene geological uplift of the Bahoruco Peninsula [34, 35], with the material likely derived from mafic igneous sources [14, 36]. Although geochemical information on these deposits is limited, they show chemical similarities with bauxites found in Jamaica and the Southwestern corner of Hispaniola (Fig 1), including the Dominican Republic and Haiti [32, 33]. Previous studies [14, 34, 35] identified REE in the inactive Las Mercedes mining site and other sites such as Aceitillar and El Turco. Through hydro-magnetic separation methods, researchers detected trace amounts of REE-bearing carbonate and phosphate minerals, representing the original form of the REE before millions of years of geochemical weathering.

From a climatic perspective, LeBlanc et al. [37] summarized the DR's overall national climate as tropical raining (class A on the Köppen [38] scale, based on its monthly mean air temperature > 18˚C and annual mean rainfall amounts), with substantial local variability caused by the Cordillera Central mountain range, which blocks the onshore flow of humid ocean air from the northeastern trade winds and creating a strong rain shadow effect (and subsequent aridity) in the southwestern portion of the country [37], particularly around the Pedernales area. The coastal areas in the Bahoruco Peninsula receive only 35–40 cm of annual rainfall, consistent with an arid steppe climate (Köppen category BShw), while the highland areas of the Sierra de Bahoruco, located 10–15 km north of Pedernales, transition to a tropical rainforest climate (Köppen category Af) at an elevation of approx. 1220 m. Apparently, this climate has persisted for millennia [39], with the Dominican Republic receiving much of its annual moisture from hurricane events, which are strongest during the La Niña phase of the El Niño Seasonal Oscillation (ENSO).

Given the location of the karstic bauxite deposits on gentler slopes and inclines throughout the Sierra de Bahoruco, we approached the exploration from a Soil Science perspective. We hypothesized that intense geochemical weathering from the local climate concentrated these

recalcitrant elements resulting in elevated concentrations (compared to background levels). Specifically, we hypothesized that the sustained, intense geochemical weathering of volcanic ash and ancient erosional/paleosol material disintegrated a significant proportion of the previously identified [35] REE-bearing phosphate (e.g., monazite-Ce, La) and carbonate minerals. As "free" ions, REE would be retained through reversible cation exchange reactions, particularly in the argillic subhorizons of the soil [9, 40]. Clay accumulation in the subsurface and the development of argillic horizons would be consistent with classical soil development, leading to the translocation of fine particles through the soil profile. However, REE-bearing mineral particulates would not necessarily conform to this pattern, and may exhibit limited soil profile infiltration.

The Sierra de Bahoruco exists is located in a remote region of the Dominican Republic, along the coastal, southwestern border with Haiti. Geochemical exploration in this area presents several logistical, infrastructural, and technical challenges. Although it is possible to transport samples to developed regions of the country such as the capital, Santo Domingo, the journey takes at least six hours on partially developed mountain roads. The large number of samples required to explore the bauxite deposits would significantly increase operational costs, risks, and overall uncertainty of the campaign before any actual analytical work could begin. Therefore, we opted to focus on a portable sensor approach that would ultimately facilitate the selection of promising sites for exploratory drilling and trenching. The objectives of this study were to (i) characterize the REE concentrations in the bauxite deposits across the area and (ii) explore the viability of two different sensors to facilitate geochemical exploration of REE in remote areas.

## Materials & methods

### Field measurements and sampling

Explorations were conducted in the south-facing Sierra de Bahoruco mountains north of Pedernales, where bauxite samples were collected and field characterized from several previously identified deposits (Table 1, Fig 2A). Among the sampled sites, Aceitillar and Las Mercedes showed heavily disturbance due to prior bauxite mining. While it was unclear whether bauxite had been removed from the Aceitillar site, up to 18.3 m of material appeared to have been extracted at Las Mercedes. The remaining sites showed less disturbed, primarily from agricultural activities. Soil samples were taken from both the surface and subsurface (Fig 2B), aligned our hypothesis about REE distributions and natural soil morphological development. Sampling locations were selected based on visible differences between bauxite-rich and calcite-derived soils. The bauxite soils were reddish in appearance, gently undulating, and typically free of gravel or large rocks, while the calcite soils were rocky, sparsely vegetated, and steep, typical of karst topography. Indicators such as fruit trees and farmland further served to define the boundaries of the bauxite deposits, which were more fertile compared to the calcite soils. Larger deposits were sampled at higher and lower elevations, targeting summit or footslope landscape positions if possible. For each selected deposit, soil was dug using a posthole digger, compositing material from four different holes dug to account for soil heterogeneity. Samples were collected at the surface and then subsurface, at no more than 30 cm due to the nearly impenetrable dense clay layer. Field parameters such as soil color (Munsell color scale), texture (hand test), and, occasionally, effervescence, using a 1% HCl solution [41] were recorded. Geospatial data, elevation, landscape position, and slope were also documented. However, we could not assign full NRCS Soil Taxonomy (ST) designations [41, 42] because the dense clay subsurface made it difficult to access the full soil profile. Additionally, we were unable to find documented annual precipitation data for the region.

**Table 1.** *Samples collected by bauxite deposit group with their corresponding geospatial coordinates and elevation information.*

| Deposit group | Soil layer | Average depth (m) | Latitude | Longitude | Elevation (m) | Soil Texture | Soil Munsell color |
|---|---|---|---|---|---|---|---|
| AguasNegras | surface | 0 | 18.199 | -71.688 | 1074 | clay | 7.5YR 4/4 |
| AguasNegras | subsurface | 45.08 | | | | clay | 5YR 5/6 |
| AguasNegras | surface | 0 | 18.1899 | -71.6883 | 1074 | clay loam | 7.5YR 3/3 |
| AguasNegras | subsurface | ND | | | | clay | 2.5YR 4/4 |
| AguasNegras | surface | 0 | 18.1863 | -71.6906 | 996 | clay loam | 5YR 3/3 |
| AguasNegras | subsurface | 74.3 | | | | clay | 5YR 5/6 |
| AguasNegras | surface | 0 | 18.1847 | -71.6912 | 995 | clay | 5YR 3/4 |
| AguasNegras | surface | 0 | 18.17128 | -71.67934 | 956 | clay | 7.5YR 4/6 |
| AguasNegras | surface | 0 | 18.17127 | -71.67952 | 954 | clay loam | 2.5YR 4/4 |
| AguasNegras | subsurface | 55.88 | | | | clay | 2.5YR 3/6 |
| AguasNegras | surface | 0 | 18.17128 | -71.68122 | 956 | clay loam | 2.5YR 3/4 |
| AguasNegras | subsurface | 50.8 | | | | clay | 2.5YR 5/4 |
| AguasNegras | surface | 0 | 18.1723 | -71.6759 | 952 | clay | 5YR 4/4 |
| AguasNegras | subsurface | 50.8 | | | | clay | 2.5YR 4/4 |
| AguasNegras | surface | 0 | 18.171842 | -71.676528 | 982 | clay | 5YR 4/3 |
| AguasNegras | subsurface | 48.26 | | | | clay | 5YR 3/4 |
| Altagracia | surface | 0 | 18.1946 | -71.7019 | 999 | clay loam | 2.5YR 2.5/4 |
| Altagracia | subsurface | 67.31 | | | | clay | 10R 3/4 |
| Altagracia | surface | 0 | 18.1944 | -71.7019 | 1011 | clay | 5YR 4/4 |
| Altagracia | subsurface | 80.64 | | | | clay (clay increase) | 5YR 3/4 |
| Altagracia | surface | 0 | 18.1903 | -71.6988 | 1012 | clay | 5YR 3/3 |
| Altagracia | subsurface | 71.12 | | | | clay (clay increase) | 5YR 3/4 |
| Altagracia | surface | 0 | 18.192 | -71.6965 | 1012 | clay | 2.5YR 3/4 |
| Altagracia | subsurface | 66.04 | | | | clay (clay increase) | 2.5YR 5/4 |
| Altagracia | surface | 0 | 18.1903 | -71.7004 | 978 | clay | 5YR 3/4 |
| Altagracia | subsurface | 56.52 | | | | clay | 5YR 3/3 |
| Avila | surface | 0 | 18.1391 | -71.7007 | 552 | clay loam | 2.5YR 5/4 |
| Avila | subsurface | 52.7 | | | | clay | 2.5YR 5/4 |
| Avila | surface | 0 | 18.1372 | -71.698 | 564 | clay loam | 2.5YR 4/4 |
| Avila | subsurface | 59.69 | | | | clay | 2.5YR 5/6 |
| Avila | surface | 0 | 18.1378 | -71.6985 | 545 | clay | 2.5YR 3/4 |
| Avila | subsurface | 65.4 | | | | clay (clay increase) | 2.5YR 2.5/4 |
| Avila | surface | 0 | 18.1466 | -71.6959 | 638 | clay | 2.5YR 2.5/4 |
| Avila | subsurface | 57.15 | | | | clay | 10R 3/4 |
| Avila | surface | 0 | 18.1442 | -71.6943 | 648 | clay loam | 2.5YR 2.5/4 |
| Avila | subsurface | 53.98 | | | | clay | 2.5YR 3/6 |
| Avila | surface | 0 | 18.144 | -71.6934 | 641 | clay loam | 2.5YR 3/4 |
| Avila | subsurface | 57.15 | | | | clay | 10R 3/6 |
| Avila | surface | 0 | 18.139 | -71.694 | 589 | clay | 2.5YR 3/4 |
| Avila | subsurface | 50.16 | | | | clay | 10R 3/4 |
| Avila | surface | 0 | 18.15775 | -71.707302 | 665 | clay loam | 7.5YR 3/4 |
| Avila | subsurface | 73.66 | | | | clay | 7.5YR 3/4 |
| Avila | surface | 0 | 18.147951 | -71.704788 | 648 | clay loam | 2.5YR 2.5/3 |
| Avila | subsurface | 68.58 | | | | clay | 2.5YR 3/6 |
| Guerrero | surface | 0 | 18.1254 | -71.6763 | 570 | clay | 2.5YR 2.5/4 |
| Guerrero | subsurface | 59.06 | | | | clay | 2.5YR 2.5/4 |
| Guerrero | surface | 0 | 18.125985 | -71.644119 | 727 | clay | 2.5YR 4/4 |

(*Continued*)

**Table 1.** (Continued)

| Deposit group | Soil layer | Average depth (m) | Latitude | Longitude | Elevation (m) | Soil Texture | Soil Munsell color |
|---|---|---|---|---|---|---|---|
| Guerrero | subsurface | 73.66 | | | | clay | 2.5YR 3/6 |
| LaAltagracia | surface | 0 | 18.121354 | -71.599003 | 1108 | clay loam | 2.5YR 2.5/4 |
| LaAltagracia | subsurface | 55.88 | | | | clay | 2.5YR 3/6 |
| LasMercedes | surface | 0 | 18.0838 | -71.6539 | 412 | clay loam | 2.5YR 2.5/4 |
| LasMercedes | subsurface | 55.24 | | | | clay loam | 2.5YR 2.5/4 |
| LasMercedes | surface | 0 | 18.083849 | -71.650496 | 435 | clay loam | 2.5YR 2.5/4 |
| LasMercedes | subsurface | 99.06 | | | | clay | 2.5YR 2.5/4 |
| LosArroyos | surface | 0 | 18.2348 | -71.7521 | 1190 | clay | 5YR 3/4 |
| LosArroyos | subsurface | 57.79 | | | | clay (clay increase) | 2.5YR 3/6 |
| LosArroyos | surface | 0 | 18.2341 | -71.7523 | 1176 | clay loam | 5YR 4/4 |
| LosArroyos | subsurface | 55.88 | | | | clay | 2.5YR 4/6 |
| LosArroyos | surface | 0 | 18.2327 | -71.7536 | 1151 | clay | 5YR 4/6 |
| LosArroyos | subsurface | 62.23 | | | | clay (clay increase) | 2.5YR 4/6 |
| LosArroyos | surface | 0 | 18.244414 | -71.750804 | 1363 | clay | 5YR 4/6 |
| LosArroyos | subsurface | 73.66 | | | | clay | 5YR 4/6 |
| Mango | surface | 0 | 18.1048 | -71.7178 | 253 | clay loam | 5YR 3/4 |
| Mango | subsurface | 47.62 | | | | clay | 2.5YR 3/4 |
| Mango | surface | 0 | 18.10311 | -71.72207 | 252 | clay loam | 5YR 3/4 |
| Mango | subsurface | 43.18 | | | | clay loam | 5YR 3/4 |
| Mango | surface | 0 | 18.10347 | -71.72255 | 242 | clay loam | 5YR 3/4 |
| Mango | subsurface | 73.66 | | | | clay loam | 2.5YR 2.5/4 |
| SitiosQuemado | surface | 0 | 18.1377 | -71.6884 | 587 | clay | 2.5YR 3/4 |
| SitiosQuemado | subsurface | 60.33 | | | | clay | 2.5YR 3/4 |
| SitiosQuemado | surface | 0 | 18.1297 | -71.6877 | 552 | clay loam | 2.5YR 2.5/4 |
| SitiosQuemado | subsurface | 63.5 | | | | clay | 10R 3/6 |
| Yagrumo | surface | 0 | 18.1034 | -71.6763 | 450 | clay | 2.5YR 2.5/4 |
| Yagrumo | subsurface | 76.84 | | | | clay | 2.5YR 3/6 |
| Yagrumo | surface | 0 | 18.103275 | -71.67592 | 483 | clay loam | 2.5YR 3/4 |
| Yagrumo | subsurface | 74.93 | | | | clay | 2.5YR 3/6 |

## General soil characterization

Collected samples were air-dried, ground, and sieved (2 mm) in Pedernales before being shipped to the U.S. Army Engineer Research & Development Center (ERDC) in Vicksburg, MS, USA, for laboratory analysis. The "bulk" composition [43–45] of soil elements (including REE) was analyzed using microwave-assisted acid digestion and inductively coupled plasma with an Agilent 5110 optical emission spectroscopy (ICP-OES). For each sample, 0.5 g was digested (in triplicate) using EPA Method 3051 [46] with a Mars 6 (CEM, Matthews, NC, USA) OneTouch microwave digestor, followed by diluting (to approx. 40 mL) and filtering (using 0.45 μm syringe filters). Soil carbon speciation was determined with an Elementar Soli-TOC instrument (Elementar Americas, Ronkonkoma, NY, USA).

## pXRF analysis

Portable x-ray fluorescence (pXRF) analysis was performed using a Thermo Niton5 XL Plus handheld spectrometer. Preliminary experiments showed that using the pXRF in the field without sample preparation typically underreported both the presence and concentration of REE in the soils–a phenomenon commonly attributed to irregular particle size and the soil

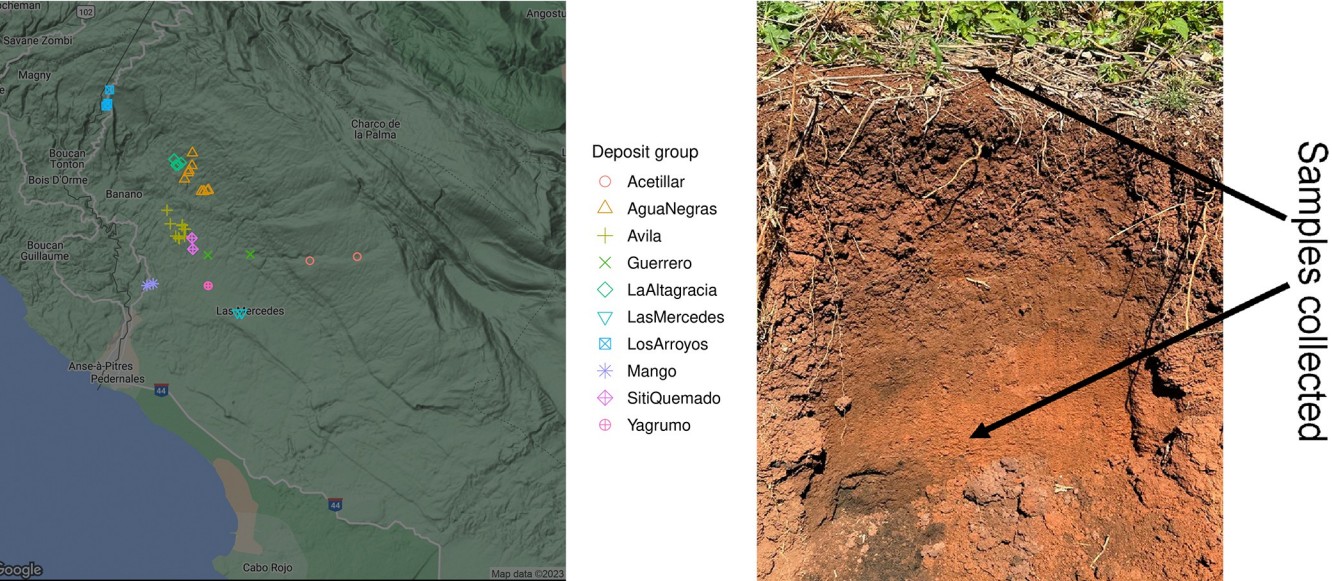

**Fig 2.** (A) Map showing the distribution of sampling sites; (B) Image showing natural horizonation of the bauxite deposits from a soil profile cutout created by the landowner. Samples were collected at the surface and the subsurface, with the strong color change, indicating the natural boundary between the typical clay loam A horizon and the argillic (clay) subsurface horizons ($Bt_1$ and $Bt_2$).

moisture. [47] Thus, the air-dried samples and ground samples were powderized using a coffee grinder to pass through a 0.425 mm sieve, and analyzed using the manufacturer's proprietary method for REE that measured, in total 44 elements including 6 REE. The method employs four different filters (called main, low, high, and light). We set our collection times at 15 s for each filter, which we deemed the most efficient, given the energy fluctuations observed for longer dwell times. The instrument's internal calibration was checked daily using the USZ 42 REE Ore PP 180–654 and NIST2709a pp (180–649) standards. The optimal x-ray fluorescence signal was obtained by placing the spectrometer directly in a plastic weigh boat filled with the powdered sample. Given the microscale heterogeneity of elemental domains commonly manifested in processed soil samples [43, and references therein], measurements were repeated ten times on each sample, with the detector repositioned at a different place in the weight boat for each measurement.

## Potentiometric measurements

Soil pH, electrical conductivity (EC), and redox potential (Eh) were measured using a battery-operated, Hach (Hach Company, Loveland, Colorado, United States) sensION+ MM150 portable multi-parameter meter and field kit with multi-sensor combination probe, calibrated with the manufacturer's proprietary standards.

## Soil color measurements

Soil "color" was determined on the soils using a self-calibrating NixPro2 color sensor (Nix Sensor, Ltd., Hamilton, Ontario, Canada). The NixPro2 operates as a tristimulus colorimeter, measuring color based on three primary color components rather than capturing a full spectral range. Consequently, the instrument does not possess a specified wavelength range similar to a spectrophotometers. Instead, it utilizes high-CRI (Color Rendering Index) LEDs with color temperatures of 5000K and 6500K to illuminate the sample, ensuring accurate color

measurement under standardized lighting conditions. Using this sensor, measurements were collected on the processed soils given the sensitivity of soil color to moisture. Color measurements were determined by gently pushing the sensor's collection chamber beneath the sample surface and averaging the results of three different scans for different sample positions in the weigh boat.

## Computational modeling and data visualizations

Statistical analyses, including boxplots and geometric transformations were performed using the R programming language [48] via the RStudio interface [49]. Most visualizations were created using ggplot [50] while ternary diagrams were created using the ggtern package [51] in R. Compositional data analysis (CoDa) was used for the bulk and pXRF datasets to correct for geometric distortions in proportional data [52–55]. Variables containing > 5% missing data (e.g., values below the detection limit) were removed, and remaining left-censored were imputed using the log-ratio Expectation-Maximization (lrEM) algorithm within the zCompositions package [56]. Next, each row was closed to a unit sum 100 using the *clo* command [57]. Finally, all rows for the bulk soil characterization data (from here termed the bulk composition) were transformed using the centered log-ratio *(clr)* transformation as reported previously [43–45] while the pXRF data were transformed using the additive-log ratio *(alr)* transformation as follows (Eq 1):

$$alr(x) = \left[ \ln\left(\frac{x_1}{x_{Si}}\right), \ldots, \ln\left(\frac{x_{D-1}}{x_{Si}}\right) \right] \qquad [1]$$

where D is the number of components, or features, in the composition. Eq 1 shows that all variables were normalized to Si. This was done to calibrate the pXRF response as Si was dropped from the bulk composition given that quartz is not quantitatively consumed in the acid digest method. The transformed bulk composition dataset consisted of 25 (D-1) components across 162 samples while the transformed pXRF composition comprised 41 components and 749 samples (considering the ten replicate scans per sample described above). The color sensor dataset (being noncompositional was not transformed) was comprised three components and 65 samples. Handling left-censored data was particularly problematic for the pXRF composition since many elemental concentrations were below detectable levels. However, the handheld XRF uses a proprietary Fundamental Parameter algorithm to estimate elemental concentrations, and the instrument reports the uncertainty of these estimates as two standard deviations [58]. If the uncertainty was greater than the detection limit, the uncertainty value was considered the maximum concentration the XRF could have detected. Thus, the uncertainty values (2 - σ) were added to the detected values to create a new pXRF compositional dataset. The remaining left-censored data were processed using the method described above.

The final preprocessing step before using the data for ML modeling was to associate the target REE data with the composition data. Both the bulk and pXRF compositions were collected as replicates, with three replicates per sample for the bulk composition and ten replicates per soil sample for the pXRF. Only the median value of each replicate set was used for modeling, as the median is considered a more accurate representation of the measurements compared to the mean.

## Geochemical clustering of samples

Statistical clustering of the CoDA-transformed geochemical data was performed using the nonlinear dimension reduction technique, Uniform Manifold Approximation and Projection or UMAP [59] via the "uwot" package [60] for R. A supervised UMAP model was generated

**Table 2.** *List of candidate models and hyperparameters used for preliminary model selection.* A range of values was assigned for each hyperparameter. The Scikit-Learn 'GridSearch_CV' function, was used to create a grid of all possible combinations of these hyperparameters. Cross-validation was then applied to evaluate model performance for each combination.

| Algorithm | Hyperparameter 1 | Hyperparameter 2 | Hyperparameter 3 |
|---|---|---|---|
| SVM | C = $1e^3$, $1e^4$ | gamma = $1e^{-3}$, $1e^{-2}$, "scale" | kernel = rbf, poly, linear |
| Extra Trees | n_estimators = 100, 300 | min_samples_leaf = 1, 2 | max_features = sqrt, log, None |
| Random Forest | n_estimators = 100, 300 | min_samples_leaf = 1, 2 | max_features = sqrt, log, None |
| XGBoost | n_estimators = 100, 200 | eval_metric = mlogloss, merror | booster = gbtree, gblinear |
| K-Nearest Neighbors | n_neighbors = 1, 2, 3 | weights = uniform, distance | p = 1, 2 |

based on the assigned sample deposit groups, with hyperparameters n_neighbors and min_dist hyperparameters set at 25 and 0.2, respectively.

## Machine learning (ML) regression modeling

ML modeling was performed using Python3. The preprocessed CoDA data, as described earlier, was used to train and test the models. Initial model selection involved training a suite of candidate regression models using 3-fold cross-validation on the pXRF composition dataset to predict the sum of REE (ΣREE). The algorithms considered included were two instance-based models, Support Vector Machine (SVM) and K-Nearest Neighbors (KNN), four tree ensemble models, Random Forest (RF), Extremely Randomized Trees (ET) [61], XGBoost (XGB) [62] and the ordinary least squared (OLS) regression model. Each of these ML models approaches data pattern recognition differently, allowing for a variety of perspectives when modeling REE in bauxite. The 'GridSearchCV' function from the *scikit-learn* library was used to explore a grid of hyperparameters among each algorithm [63], listed in Table 2. Among these models, SVM, RF, and ET models all performed equally well, but the ET model was selected for further development due to our prior experience with this algorithm in modeling soil geochemistry [43].

After selecting the ET model, a resampling procedure was performed to assess the generalization accuracy of the model across the three datasets (bulk composition, pXRF, and color sensor data). This resampling involved generating 500 randomized train-test splits (70% - 30%), where the model was refit at each iteration. Given the wide range of target variable values (ΣREE), binned stratification was applied to ensure balanced sampling across the different ranges of REE concentrations [64]. Each sample was assigned into one of five bins based on its ΣREE value, and an equal proportion from each bin was included in each of the 500 train-test splits. Upon completion of the 500 iterations, the model with the lowest root mean squared error (RMSE) for ΣREE was selected as the final (optimized) model. In addition, the average RMSE over 500 models was calculated as a secondary estimate of model performance to provide a realistic measure of the model's generalization accuracy.

## Visible-Near-Infrared (vis-NIR) spectroscopy

Preliminary spectra of processed bauxite samples were collected using an ASD (Analytical Spectral Devices, Worcestershire, UK), FieldSpec 4 using a contact probe. Five scans were averaged, baseline corrected, smoothed via a Savitzkty-Golay filter, and SNV normalized. The first derivative of the treated spectra was calculated in order to identify the chemical domains included the spectral response.

All notebooks and data developed for this work can be found at the following: https://github.com/candros/DR_REE_pXRF_Modeling.

## Results

### Soil profile observations

Due to the challenges of hand-digging in these soils, only limited pedomorphic descriptions were provided. With the exception of a profile cutout previously excavated by the landholder (Fig 2B), most soil profiles were summarized based on the portions accessible during field-work. Generally, the profiles consisted of an A horizon with a subangular blocky structure, typically ranging from clay loam to clay in texture. The average depth for subsurface samples was 24.5 ± 4.7 cm, with a range from 17 to 39 cm, largely determined by the depth at which we encountered the dense, impenetrable clay layer.

### Geochemistry of the different deposits

The geochemical signatures of the bauxite deposits were investigated using a compositional diagram of the soil Al, Fe, and Si, as determined from the pXRF data (Fig 3A) measured on ten different spots within each powdered sample (representing the 10 replicate measurements described earlier). The plot clearly shows different zones corresponding to the bauxite classifications established by Bardossy [15]. For example, samples from Aceitillar and portions of

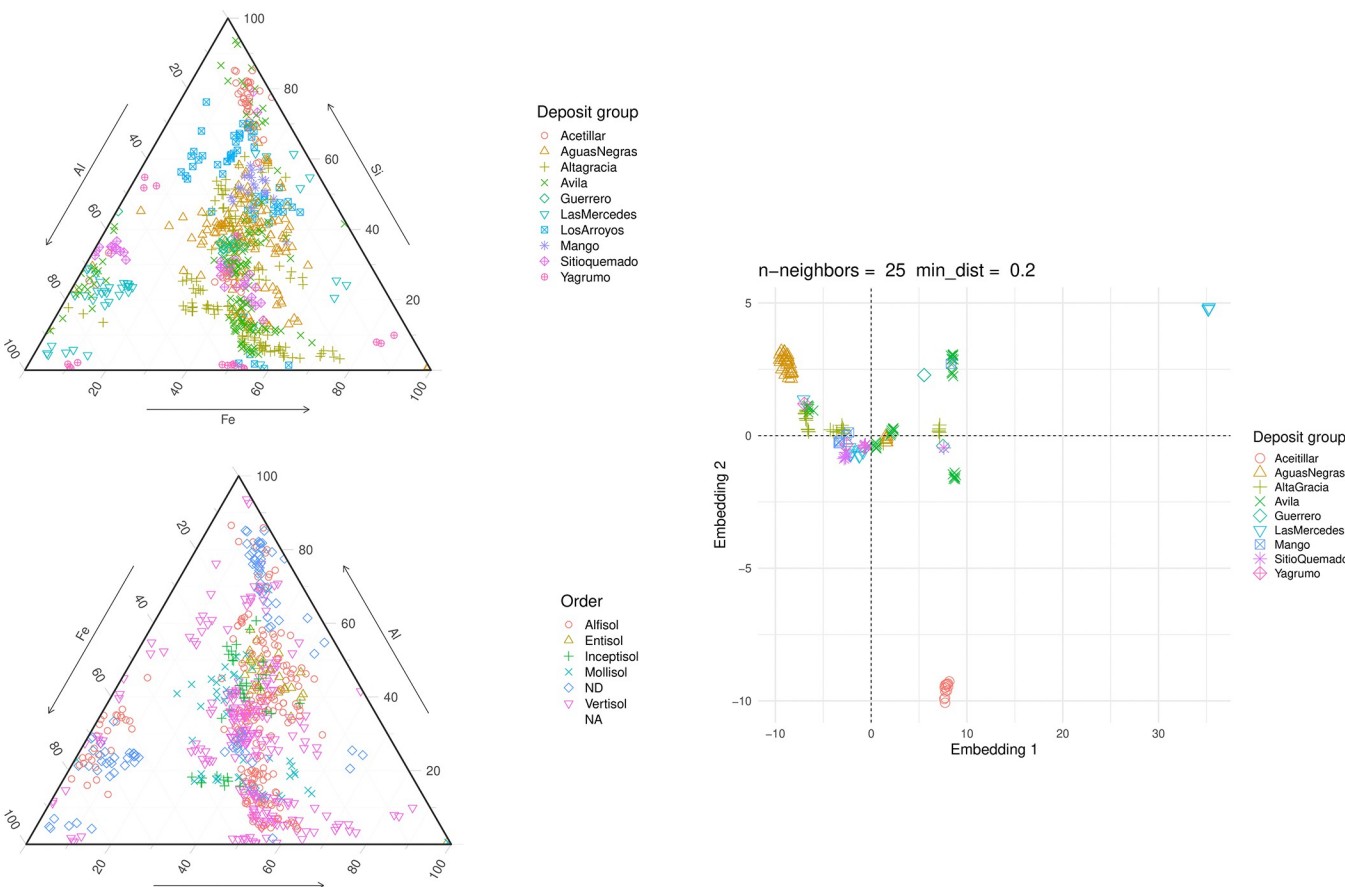

**Fig 3. Different visualizations of the geochemical data.** Ternary plots showing the relationship among Si, Al, and Fe concentrations (as determined by pXRF analysis of 10 different spots on powderized samples) with respect to (A) Deposit group and (B) Order in the NRCS soil morphological classification system. (C) UMAP decomposition of geochemical data determined by acid-digested, ICP-OES-determined REE showing the latent structural relationships among the different bauxite deposits.

Avila, consistent with the findings of Villanova-de-Benavent et al. [35], were classified as "Bauxite to Fe-rich bauxite", enriched in Al and Si, but relatively low in Fe. Los Arroyos spanned both the bauxite and Fe-rich classifications, while most samples from Mango, Aguas Negras, Guerrero, and Sitio quemado were classified as "Clayey bauxite", with relatively higher Al content compared to Si. Las Mercedes fell under "Bauxitic clayey iron ore," being enriched in Al. In contrast, samples from Avila, Alta Gracia, and Yagrumo were classified as "Bauxtic clays," characterized by relatively low Si.

When Al-Si-Fe ratios were replotted based on NRCS morphological designations (Fig 3B–note that the heavily disturbed mining sites, Aceitillar and Las Mercedes, were not morphologically characterized and not included in the plot), many of the soils with high clay content qualified as Vertisols. Two distinct clusters of Vertisols emerged: One corresponding to "Clayey bauxites" and another to "Bauxite clay." Significant overlap was observed between Alfisols and Vertisols in the ternary diagram.

Using supervised UMAP for dimension reduction of the bulk composition, (Fig 3C), we observed that deposits like Aceitillar, Aguas Negras, and certain Las Mercedes samples formed distinct clusters, were better distinguished compared to principal component analysis, multidimensional scaling, and unsupervised UMAP (S1, S2 Figs). Other deposits were less distinguishable, suggesting the need for further refinement of deposit group labels. The overlap observed may be influenced by elevation and the transition from arid to tropical climates across the Sierra de Bahoruco.

## Soil REE concentrations

The bulk composition data (plotted in $\log_{10}$ concentrations) revealed differences between surface and subsurface layers (Fig 4A). Aceitillar, a former mining site, showed the highest median surface REE concentration at 2258 mg total REE kg$^{-1}$, with one sample reaching 1.3%. Among the higher-elevation deposits (> 975 m), Los Arroyos possessed the greatest surface REE concentration (1325 mg kg$^{-1}$), about 400 mg kg$^{-1}$ higher than Avila and Guerrero. In contrast, lower-elevation deposits (≤ 610 m), such as Yagrumo, Mango, Sitio quemado, and Las Mercedes, exhibited surface REE concentrations 300 to 400 mg kg$^{-1}$ lower. At the heavily impacted Las Mercedes site, after approx. 18.3 m of bauxite removal, the median surface REE concentration was 630 mg kg$^{-1}$. In general, the subsurface was approx. 100 mg kg$^{-1}$ higher than the surface across the deposits, with variations depending on elevation.

The light-(Sc, Y, La, Ce, Pr, Nd, Sm)-to-heavy REE (Eu, Gd, Tb, Dy, Ho, Er, Tm, Yb, and Lu), LREE/HREE, ratios (Fig 4B) remained relatively constant with depth for most deposits, with some becoming more HREE-enriched at 1 m depth, except for Las Mercedes, Mango, and the Sitio quemado.

## Correlation analysis

A correlation analysis (Fig 5) revealed relationships between deposit locations and total REE concentrations. There was a strong positive correlation (r = 0.96) between the sample latitude and elevation, which makes sense given that the deposits are located on the south-facing slopes of the mountains (facing the coast), where denser vegetation tends to increase with elevation. Although exact precipitation data were unavailable, this pattern aligns with observable vegetative growth trends. Excluding the disturbed Aceitillar and Las Mercedes sites, a moderate positive correlation was found between REE concentration and both elevation (r = 0.34) and latitude (r = 0.36), suggesting higher REE concentrations occurred in areas with greater moisture. Additionally, a negative correlation between LREE/HREE ratio and both elevation (r = -0.36) and latitude (r = -0.28) was observed, indicating that the bauxites became relatively

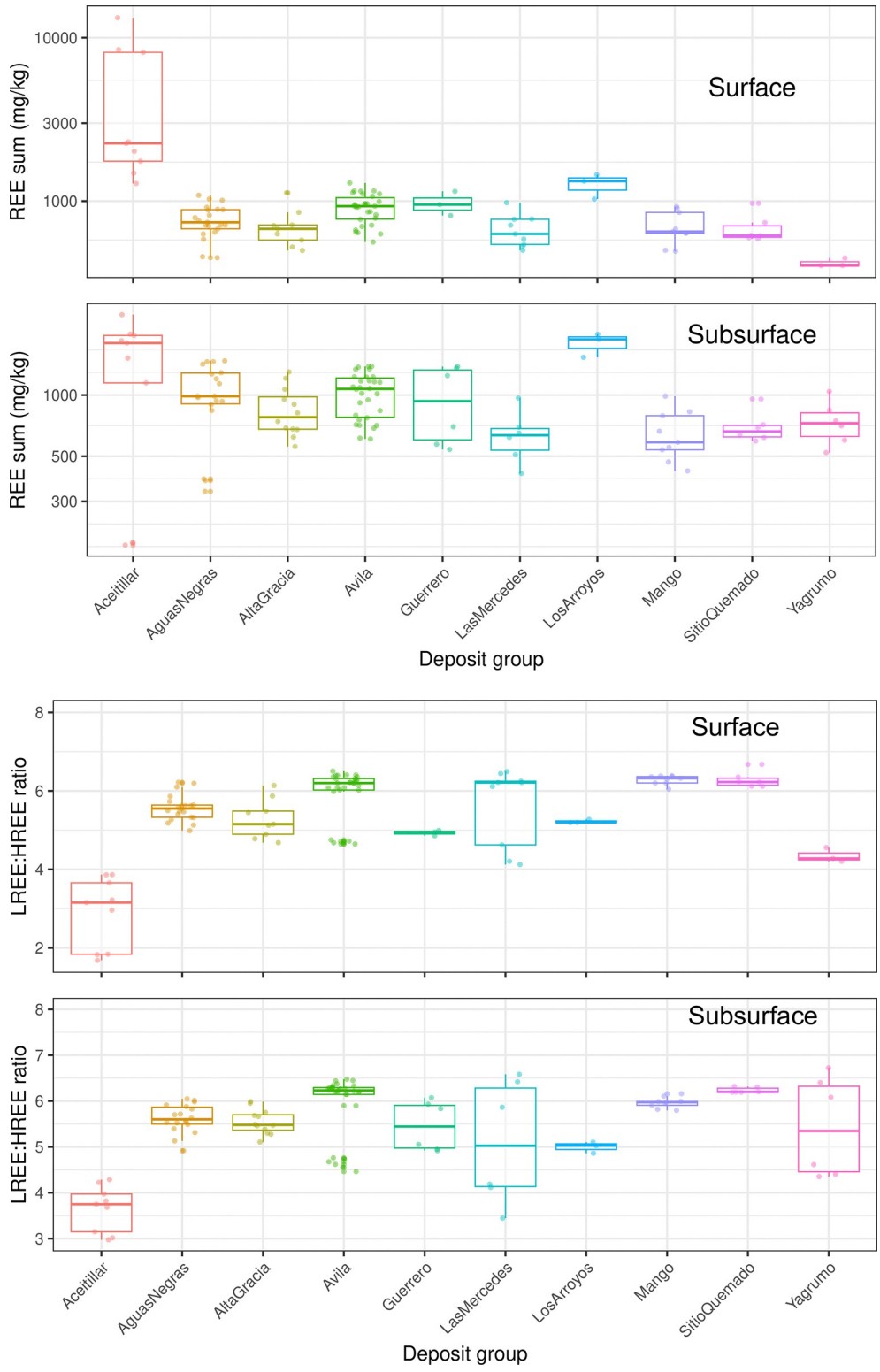

**Fig 4. Boxplots showing the statistical distributions of the REE data.** (A) the measured $\log_{10}$ REE concentration and (B) LREE/HREE ratios in the soils with respect to each particular deposit group for both the surface and subsurface depths.

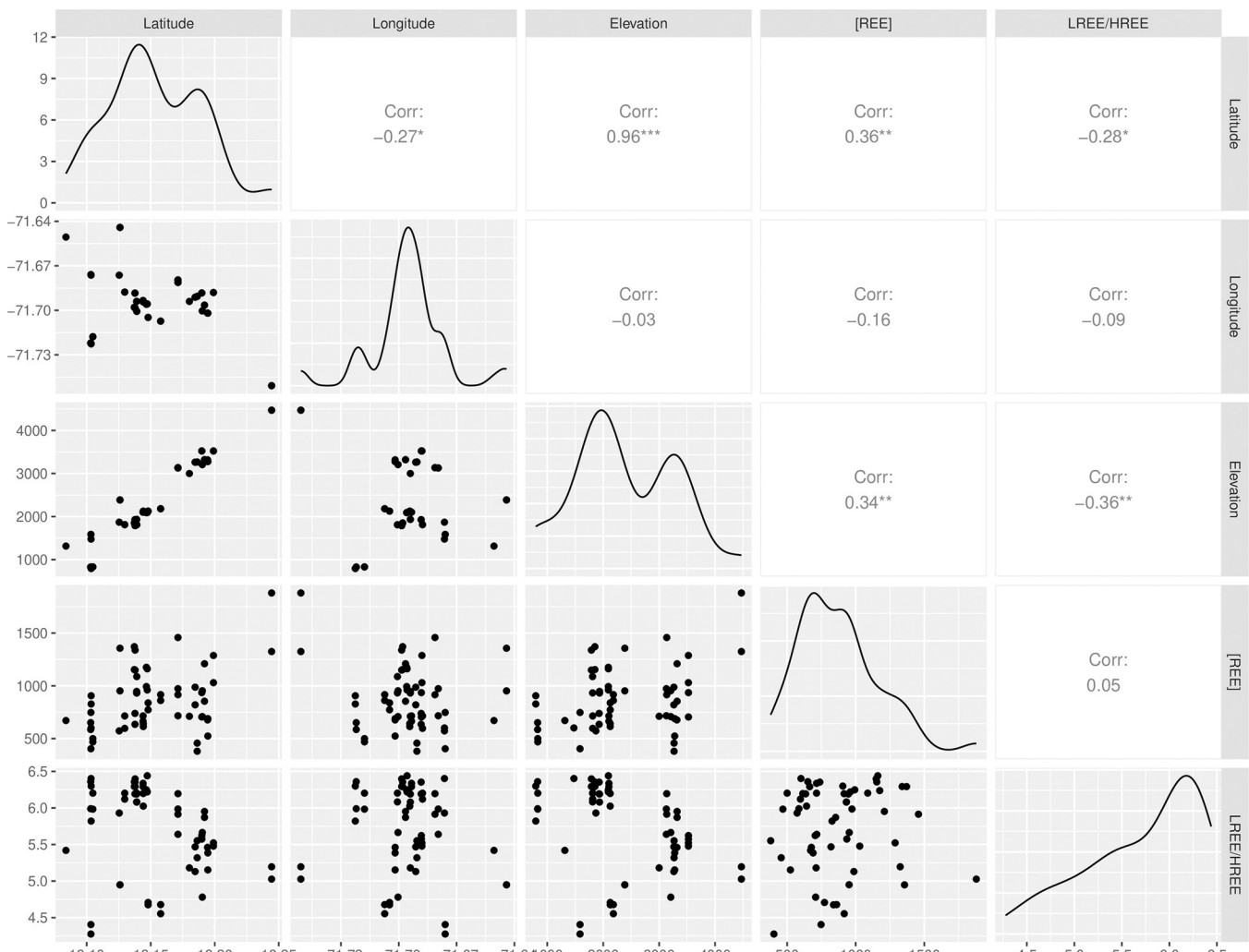

**Fig 5. Correlation plot showing the relationships between soil REE concentrations, LREE/HREE ratios, and the geographical features of sampled bauxite deposits.** Median soil REE concentrations are shown for both the surface and subsurface depths.

more enriched as the total REE concentrations increased. When the disturbed Aceitillar and Las Mercedes sites were included in the correlation analysis, the results showed a significant relationship between total REE concentration LREE/HREE ratio, and longitude, particularly influenced by the Aceitillar samples' eastern location in the Sierra de Bahuroco Mountains.

Soil pH and soil redox (Eh) values were also compared graphically (Fig 6) to illustrate the distribution of samples among the different bauxite deposits. Higher elevation samples, particularly the Los Arroyos, Avila, and Altagracia deposits, exhibited lower pH and higher Eh values. A threshold of pH 6 was used to indicate the point at which REE are expected to exist as cations (pH < 6) in soils, while soils beyond this threshold (pH > 6) suggest more complex REE bonding environment (such as mineral complexes) with solid-phase components [35].

## Calibration models for predicting soil REE concentrations

The Sierra de Bahoruco, located in the remote southwestern corner of the Dominican Republic, along the Haitian border, presents significant challenges for geochemical exploration due

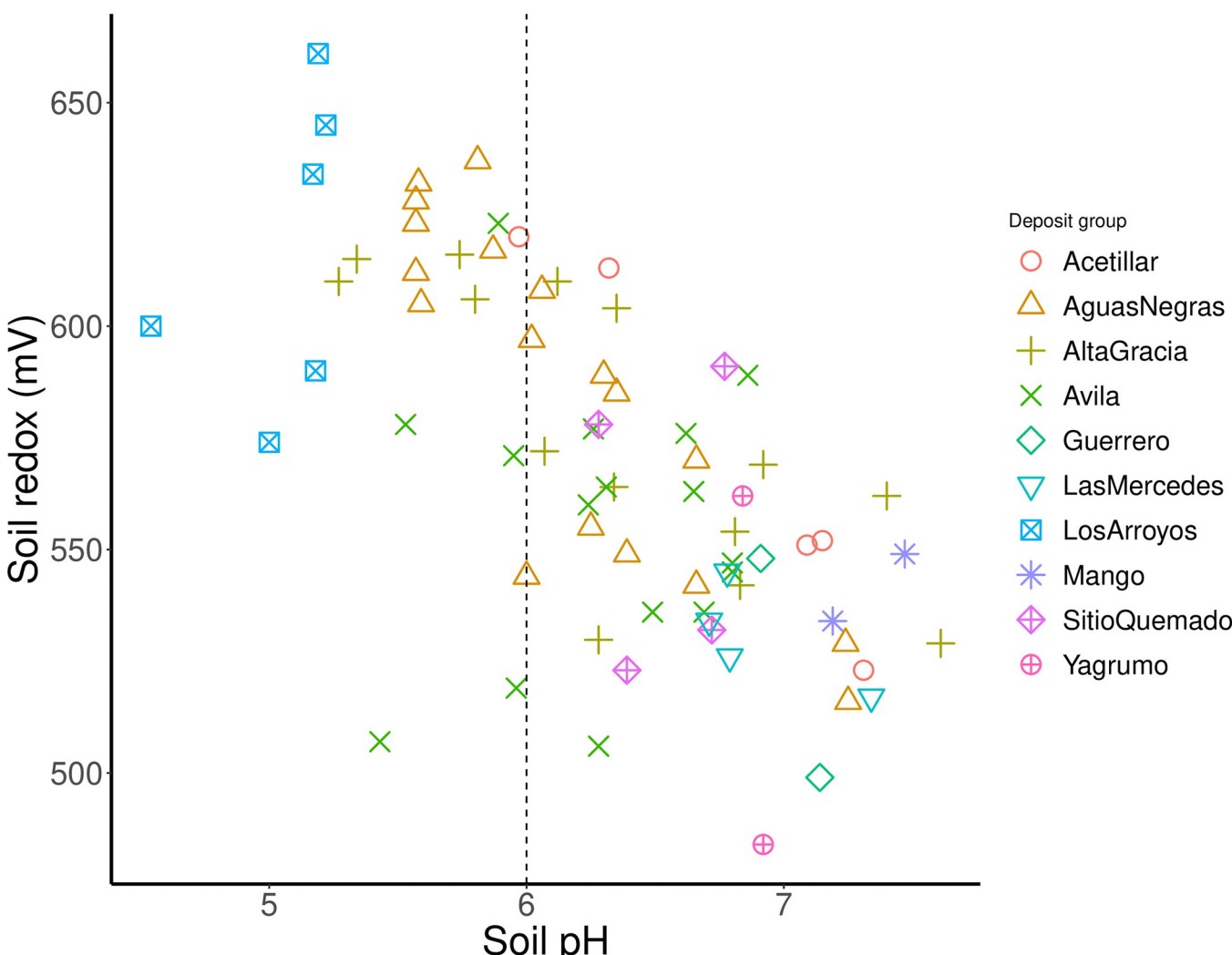

**Fig 6. Plot showing the relationship between soil pH and soil redox potential for the different bauxite deposits.**

to logistical, infrastructural, and technical limitations. While samples could be transported to more regions of the country, such as the capital, Santo Domingo, this involved a minimum six-hour drive, much of it over narrow winding roads. Given the high volume of samples required to explore the bauxite deposits, the costs and uncertainties of traditional laboratory analysis would escalate rapidly. Therefore, we focused on alternative methods for measuring the REE concentrations in this remote area.

Prediction plots generated from the optimized ET models are shown in Fig 7. As extreme geochemical outliers among the deposits, we note that the Aceitillar samples were removed from the calibration. The model based on the CoDA-based bulk composition (excluding measured soil REE concentrations) showed excellent performance, with an $R^2 = 0.96$ (Fig 7A), indicating that 96% of the variance in the observed data was explained by the model. For these models, we present the results of applying the previously described resampling method (Table 3) showing the "average" RMSE as an indicator of general model performance, accounting for variability in the test-train splits and providing a likely estimate of error in real-world applications, compared to the "optimized" RMSE representing the best-performing model

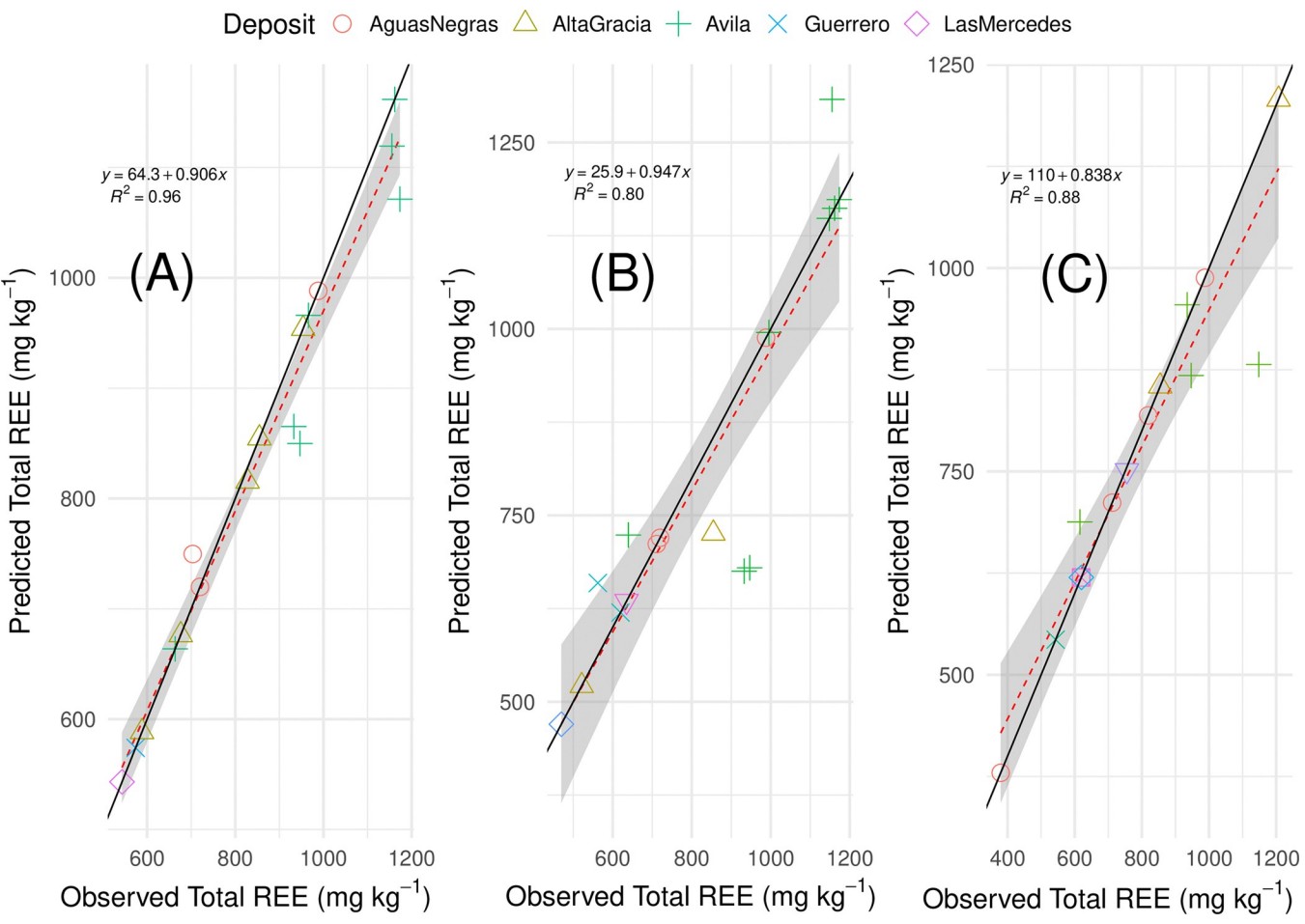

**Fig 7.** Prediction plots from the ExtraTrees regression model predicting soil REE based on (A) bulk composition, (B) pXRF analysis, and (C) NixPro2 color sensor optimized validation data. The red dotted line represents the model's function, with shading indicating the standard error. The black line represents the true diagonal of observed and predicted values.

iteration or the model's best-case scenario. While the optimized RMSE provides an optimistic view of performance that essentially establishes the upper threshold of the model's predictive accuracy, the average RMSE offers a perspective of the more realistic and probable outcome obtained from the use of this model. In view of these relationships, the bulk composition model exhibited the lowest optimized RMSE = 40.5 mg REE kg$^{-1}$, indicating excellent predictive performance in this best-case scenario. The prediction line for the bulk composition model (y = 64.3+0.906x) closely followed the diagonal, meaning the model's predictions were very accurate across the full range of observed values.

**Table 3. Generalization estimates for the soil characterization data (including REE) based on the the three sets of data (bulk composition, pXRF, and color sensor).** Root mean squared errors (RMSE) and the coefficient of determination ($R^2$) are provided for both the average and optimized test-train splits, calculating using a binned stratification resampling procedure to reduce potential bias from the train-test split in the Extremely Randomized Trees (ET) model. The values represent the results after 500 sampling iterations, illustrating both the expected (average RMSE) and best-case (optimized RMSE) model performance.

| Composition | Average RMSE | Optimized RMSE | Optimized $R^2$ |
|---|---|---|---|
| Bulk | 389.9 | 40.5 | 0.96 |
| pXRF | 416.1 | 106.8 | 0.80 |
| Color | 306.6 | 76.9 | 0.88 |

The pXRF-based model (Fig 7B) exhibiting a higher optimized RMSE (106.8 mg REE kg$^{-1}$) compared to the bulk composition, indicating larger average prediction errors. The average RMSE (416 mg REE kg$^{-1}$) suggests that under normal conditions, the model's predictions are less accurate than the bulk composition. The R$^2$ = 0.80 implies that 80% of the variance is explained, indicating reasonable but less than outstanding performance by this model. To this point, the prediction line (y = 25.9+0.947x) was slightly less aligned with the diagonal than the bulk dataset, reflecting less accuracy in the predictions, particularly for the Avila samples, reflects the challenges in maininging predictive accuracy with the pXRF at higher REE concentrations. The y-intercept = 26 mg REE kg$^{-1}$ for the optimized model represents a substantially better detection limit over previous reliable REE detection limits of 1000 mg kg$^{-1}$ [65].

The color sensor model's (Fig 7C) average RMSE is the lowest among the three (307 mg REE kg$^{-1}$), suggesting it may provide more consistent predcitions across different train-test splits. However, its optimized RMSE (77 mg REE kg$^{-1}$) is higher than the bulk model's optimized RMSE but lower than the pXRF's, indicating moderate predictive performance. The R$^2$ = 0.88 explained 88% of the data variance, demonstrating a reliable though not exceptional predictive capability. This observation is borne out in the prediction line (y = 110+0.838x), where the model is the most misaligned with the diagonal at both the low and high REE concentrations. The y-intercept (110 mg REE kg$^{-1}$) combined with the model's moderate error suggests that the color sensor struggles the most out of the three datasets with lower REE concentrations.

Balaram [66] proposed that expanding the analytical range into the NIR may capture a more complete picture of the REE content of soils. A preliminary vis-NIR analysis of one of the Aceitillar samples (Fig 8) shows that the single bands for Fe-oxyhyroxide domains (~500

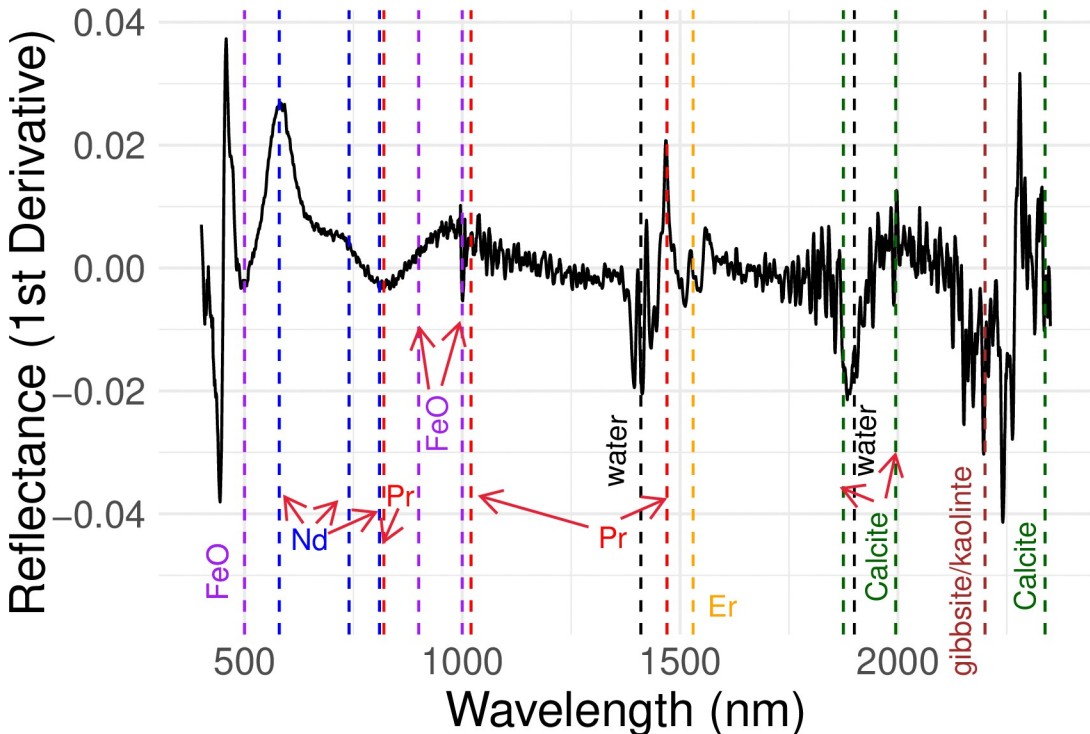

**Fig 8. Preliminary vis-NIR spectra for the Aceitillar2 sample.** The first derivative transformation of the spectra indicates the chemical domains contained within the sample based on the wavelengths of its "peaks" and "valleys".

nm) and Nd ions (~540 nm) are responding to the color sensor in the bauxite matrix, while expanding into the NIR range captures more of the Fe-O signatures as well as Pr, Er, entrained water, gibbsite/kaolinite mineralogy, and calcite.

## Discussion

From a soil science perspective [40], the results highlight differences in REE concentrations relative to the degree of material weathering and soil morphological development. Samples collected at higher elevations on the northern side of the Sierra de Bahoruco, where the climate experiences greater precipitation, generally contained higher REE concentrations. Given the widespread distribution of soils originating from the sustained weathering of volcanic ash deposited across the region, these higher soil REE are likely due to the progressive weathering at higher elevations, which concentrates elements in the soils as less resistant material dissolved away.

At these elevations, the intense weathering, represented by samples occurring with pH < 6, releases free or reversibly adsorbed REE cations (i.e., outer-sphere complexes). In this form, the REE are expected to be easily extractable via simple, less aggressive solvents. In contrast, at lower elevation sites, REE likely exist as a mix of outer- and inner-sphere complexes, requiring more aggressive extractants for their recovery. The grand exception to this theory occurs if the freed REE cations subsequently form strong complexes with the karstic bauxite's existing Fe-oxyhydroxide scavenging domains [35, 67]. In this case, the economical removal of REE could be potentially frustrated by the stability of the REE-Fe- oxyhydroxide complexes [68].

A second notable feature is the increase in REE concentrations with soil depth, where a full clay texture was commonly observed. The development of an argillic subsurface horizon, along with intense red color, indicates the translocation of fine particles and soil morphological development. This suggests that REE were similarly translocated to the deeper soil layers [67] once released from their original mineral forms. Overall, the LREE/HREE ratio was > 1, consistent with the literature on the hypothetical original pyroclastic material [33], which is enriched in minerals bearing Nd, Pr, Ce, La, and Y. However, we observed a slight decrease in the LREE/HREE ratio with depth, which aligns with the scientific literature [40] showing that the ratio can approach unity at greater soil depths.

We hypothesize the presence of a "clay lens" within a specific subsurface layer that may contain elevated REE concentrations due to natural soil development. The depth of this hypothetical clay lens, beyond the reach of our hand tools employed in this study, might explain the high REE concentrations observed at the Aceitilliar deposit group, a previously mined site where the exact depth of removed surface material removed is uncertain. Field observations suggest that the highest REE concentrations and corresponding LREE/HREE shifts occurred where approx. 3–6 m of surface material had been removed. In contrast, most REE concentrations at the highly disturbed Las Mercedes mine site were unremarkable, likely because about 18 m of surface material had been removed, possibly beyond the high REE clay lens. However, one sample taken from an area just above the deepest portion of the pit (approx. 25 m deep) had REE concentrations reaching around 1000 mg kg$^{-1}$ in some analytical replicates.

Further support for the connection between REE concentrations and soil development comes from the high accuracy of the calibration model based on the bulk soil composition. Although REE are typically considered "incompatible" or indifferent ions in natural systems, the model accurately predicted total REE concentrations based on soil constituents. This suggests that viewing REE concentrations through the lens of soil development can help connect long-term weathering and morphological processes driving local soil characteristics.

One limitation of this study is the method used for determining REE concentrations. Most studies employ a more aggressive lithium tetraborate–lithium metaborate flux fusion technique to digest the sample silica and REE-bearing minerals [14, 29, 34, 35, 69–71]. While resource constraints prevented us from using this approach, we found that the soil REE concentrations in our study were comparable to previously published values. This suggests that the acid-extraction approach effectively captured the REE outer-sphere complexes, making the results operationally relevant. Future studies could explore more detailed extractions, such as comparing acid- and flux-fusion digestions, sequential extraction extractions [72], or using solid-phase analysis like Synchrotron-based μ-XRF mapping and μ-x-ray diffraction (XRD). These techniques may be more efficient than other separation methods for recovering potentially microscopic REE-bearing minerals.

Both the x-ray and the color sensors, amplified with the ML modeling, seemed promising for aiding in the remote exploration of REE, despite their inherent limitations [65, 73, 74]. While pXRF is less expensive than transporting soil samples to Santo Domingo, residual soil moisture and non-uniform particle size significantly impacted data quality. As a result, samples were transported to the satellite laboratory before pXRF analysis. More efficient methods could be explored for use in expeditionary settings. For example, the visible color sensor model, used by our laboratory routinely for field-based morphological determinations, surprisingly provided fairly comparable reliability in predicting REE to pXRF that may possibly be improved by expanding the spectroscopic range to NIR wavelengths [66], capturing a small range of REE plus potential chemical complexation domains affected the REE speciation. Unlike pXRF, visible-NIR spectroscopy may be calibrated to account for soil moisture (such as using a pressure plate system) that shows up as large bands around 1410 and 1900 nm. Thus, the combination of field-portable color sensor and/or vis-NIR reflection probe and soil moisture sensors could enable *in-situ* exploration directly onsite. The comparability in the predictive quality of the different sensors used in the study is particularly compelling given that the pXRF is on the order of 1000x more expensive than the color sensor.

Overall, this paper highlights the power of ML modeling to substantially improve the value of the signal obtained by common remote sensing techniques. ML showed great promise for pushing the bounds of different sensors beyond their classical limitations and toward a quantitative solution in REE geochemical exploration. However, we demonstrate in this paper the need to utilize ML judiciously in view of and in concert with the particular features of each sensor. For example, the sensor should possess some nascent ability to directly detect REE, or at least in the sense of the color sensor, reasonably infer its presence based on the soil's morphological development. In this case, one should possess confidence in the sensor's use as a non-ML-enhanced screening tool. Otherwise, the prediction is based solely on the latent structure in the data without any actual relevance to detecting REE.

We have repeatedly emphasized the value of CoDA for dealing with the sophisticated multivariate structure existing in soil geochemistry data [43–45]. CoDA is uniquely suited to prevent drawing misleading and spurious conclusions from data. Here, we state emphatically that CoDA transformation is an absolutely essential step in preparing the data for ML modeling, particularly for the more exotic algorithms. By nature, CoDA provides the theoretical framework justifying as well as requiring the worker to make strategic decisions about how the data will be collected and prepared (e.g., what data to keep, what to toss out, setting detection limit thresholds, etc.), organized (such as in separate compositions), and transformed (e.g. the log-ratio routine used). Such requirements are often more "painful" than conducting the transform itself, which, in our view, serves as a barrier to the technology's wider implementation. But, in our experience, persevering to implement CoDA has always benefitted model prediction accuracies on complex soil data.

Additionally, we experimented with an approach to illustrate the best-case performance for a ML model, based on the available data, as represented by the reported optimized and average RMSE values. This disparity between these two conditions was most pronounced in the case of the bulk composition dataset, where the average RMSE was nearly nine times greater than the optimized RMSE. This gap likely resulted from the limited size of the training dataset, as a larger training set would presumably reduce the difference between the performance of the best-performing train-test split (optimized RMSE) and the average performance across all splits (average RMSE). This observation underscores the need for caution when relying on a single train-test partition to evaluate model performance. Even when partitions are randomly generated and applied to a reasonably balanced dataset, model performance can vary significantly depending on the specific partition used. This variability emphasizes the importance of employing resampling techniques like the one used in this study, or alternatively, cross-validation, to obtain a more reliable and realistic measure of a machine learning model's generalization accuracy. Overall, this analysis can be useful in assessing the quality and quantity of data collected during any potential exploration, in statistically determining whether further investigation and data collection are required.

Setting aside external factors related to the location and physical accessibility of the bauxite deposits, the eventual economic feasibility of exploiting these deposits in an environmentally feasible manner is controlled both by REE concentration and speciation/complexation. In our opinion, the true value of the pXRF and color sensor technologies would be realized if their response can be calibrated against REE speciation either alone or in combination with other sensing, analytical, and algorithmic techniques. With substantial effort, REE speciation can be determined in the laboratory using a combination of approaches, but we are not aware of any studies seeking to apply these approaches for portable sensors.

## Conclusions

This study provided valuable insights into the geochemical distribution of REE in the karstic bauxite deposits of the Sierra de Bahoruco region, Dominican Republic. By focusing on soil development, we gained a clearer understanding of the REE concentrations, light-to-heavy REE ratios (LREE/HREE), and their vertical and lateral distribution in the region. Given the remote location of the Pedernales area and the limited infrastructure, the use of pXRF, the color sensor, and other expeditionary analytical techniques could prove helpful in screening for potentially promising sites to explore for the resource assessment. When rigorously calibrated against laboratory data, these tools can serve as an effective and cost-efficient sensors for remote REE geochemical exploration, particularly in areas lacking full laboratory infrastructure. However, caution is necessary when using these sensors, as their accuracy may decrease in geochemically diverse regions, such as containing pronounced calcite intrusions, which would require separate model calibrations to main accuracy.

The analysis revealed that REE concentrations increased with soil depth, and at higher elevations that experience greater precipitation. This suggests that progressive weathering processes in these highland areas contributes to the higher concentration of REE especially in the deeper argillic soil horizons. Furthermore, the findings indicate that REE in the Sierra de Bahoruco region may be more easily extractable in higher-elevation areas where weathering processes are more advanced, assuming secondary REE-Fe-oxyhydroxy complexation is limited. For this reason, future work should focus on developing models that can predict bulk REE speciation based on the response of these portable sensors.

Viewing the REE concentrations, LREE/HREE ratios, and their distribution across different deposits from a soil developmental perspective was instructive for understanding the state and

properties of the resource in the Sierra de Bahorcuo region, particularly given the suspected surficial nature of these deposits. Considering the current remoteness of the Pedernales area and the rugged terrain in which most of the deposits are located, expeditionary techniques and approaches that are cost-effective and robust despite limited laboratory infrastructure, were essential for conducting the REE geochemical exploration. Thus, x-ray and color sensors and possibly other analytical techniques, may be useful for advancing exploration in even more remote areas, assuming the developed models are applied only to the relatively homogeneous bauxite material. For regions with calcite, new models would need to be developed to account for compositional differences.

## Supporting information

**S1 Fig. Biplots of generated from a robust PCA for the CoDA-transformed pXRF data.**
Samples were group based on (A) deposit and (B) depth of collection (where
superficie = surface and profundo = subsurface).
(PDF)

**S2 Fig. Statistical tests for distinguishing the different bauxite depostis.** (A) Example statistical distribution testing between two different bauxite deposit groups. (B) Preliminary multidimensional scaling (MDS) results showing the statistical clustering of the different deposit groupings.
(PDF)

**S1 File.**
(DOCX)

## Acknowledgments

The use of trade, product, or firm names in this report is for descriptive purposes only and does not imply endorsement by the U.S. Government. The tests described and the resulting data presented herein, unless otherwise noted, were obtained from research conducted under the Environmental Quality Technology Program of the US Army Corps of Engineers by the U.S. Army Engineer Research and Development Center (ERDC). Permission was granted by the Chief of Engineers to publish this information. The findings of this report are not to be construed as an official Department of the Army position unless so designated by other authorized documents. This work was partially supported by the U.S. Department of State's Embassy Science Fellowship program. The authors express gratitude to Mr. Alexander (Ted) Bryant and Ms. Juanita Aguirre of the U.S. Embassy in Santo Domingo, U.S. Department of State for facilitating the bilateral coordination between the Dominican Republic and United States government agencies and technical teams. Also, the authors express their gratitude to Ramón Morrobel and José Rodriguez from Ministerio de energía y minas, República Dominicana, and Jesus Rodriguez from Servicio geológico nacional, República Dominicana for their vital site and logistical support to field and laboratory operations within the Dominican Republic.

## Author Contributions

**Conceptualization:** Mark Chappell, Charles Andros, Autumn Acree, Yoko Masue-Slowey, Wesley Rowland.

**Data curation:** Mark Chappell, Harold Rojas, Charles Andros, Autumn Acree, Yoko Masue-Slowey, Wesley Rowland.

**Formal analysis:** Mark Chappell, Charles Andros.

**Funding acquisition:** Marcelo Salles.

**Investigation:** Mark Chappell, Harold Rojas, Christine Young, Paige Fowler, Elizabeth Lotufo, Wesley Rowland, Michelle Wynter, Leopoldo Gonzalez.

**Methodology:** Mark Chappell, Charles Andros, Autumn Acree, Yoko Masue-Slowey, Wesley Rowland.

**Project administration:** Mark Chappell, Marcelo Salles.

**Supervision:** Mark Chappell, Harold Rojas.

**Validation:** Christine Young, Michelle Wynter.

**Writing – original draft:** Mark Chappell.

**Writing – review & editing:** Charles Andros, Autumn Acree, Yoko Masue-Slowey, Wesley Rowland.

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
