## [Decision Letter · Decision Letter 0]

27 May 2024

PONE-D-23-43088Geochemical exploration of rare earth element resources in highland karstic bauxite deposits in the Sierra de Bahoruco, Pedernales District, Southwestern Dominican RepublicPLOS ONE

Dear Dr. Chappell,

Thank you for submitting your manuscript to PLOS ONE. After careful consideration, we feel that it has merit but does not fully meet PLOS ONE’s publication criteria as it currently stands. Based on the reviewer's comments  I have recommended a major revision for the article.Therefore, we invite you to submit a revised version of the manuscript that addresses the points raised during the review process.

1. The introduction section currently occupies an excessive proportion of the entire paper. It includes numerous general statements that are widely known. To improve the paper, condense the introduction by focusing on specific background information pertinent to your study and removing overly general statements.

2. Please add a section on regional geology and tectonics after the introduction. This section should provide detailed context on the geological and tectonic setting relevant to your study area.

3. Ensure consistent use of the term "REE" throughout the manuscript. This will enhance clarity and avoid confusion.

4. The methodology for the machine learning predictive modeling lacks sufficient detail. Enhance this section by providing a thorough explanation of the methods used, including algorithms, parameters, and any preprocessing steps. Please add some supplementary files related to ML data.

5. Add a discussion on the implications of your study. Explain how your findings impact the field and suggest potential applications or directions for future research.

6. Strengthen the conclusion section. Summarize the key findings of your study clearly and concisely. Highlight the significance of your results and how they contribute to the existing body of knowledge.

7. Proofread the whole manuscript and make sure it is consistent and in a logical flow. 

We look forward to receiving your revised manuscript.

Kind regards,

Rizwan Sarwar Awan

Academic Editor

PLOS ONE

Additional Editor Comments:

Comments from PLOS Editorial Office: We note that one or more reviewers has recommended that you cite specific previously published works. As always, we recommend that you please review and evaluate the requested works to determine whether they are relevant and should be cited. It is not a requirement to cite these works. We appreciate your attention to this request.

Reviewers' comments:

Reviewer's Responses to Questions

**Comments to the Author**

1. Is the manuscript technically sound, and do the data support the conclusions?

Reviewer #1: Yes

Reviewer #2: Yes

Reviewer #3: Yes

2. Has the statistical analysis been performed appropriately and rigorously? 

Reviewer #1: Yes

Reviewer #2: No

Reviewer #3: Yes

3. Have the authors made all data underlying the findings in their manuscript fully available?

Reviewer #1: Yes

Reviewer #2: Yes

Reviewer #3: No

4. Is the manuscript presented in an intelligible fashion and written in standard English?

Reviewer #1: No

Reviewer #2: Yes

Reviewer #3: Yes

5. Review Comments to the Author

Reviewer #1: RE: Geochemical exploration of rare earth element resources in highland karstic bauxite deposits in the Sierra de Bahoruco, Pedernales District, Southwestern Dominican Republic

Using different geochemical techniques, this manuscript provides good information about the geochemical discoveries of rare earth element reserves in karst bauxite deposits in the Sierra de Bahoruco, Pedernales District, Southwestern Dominican Republic. Overall, the manuscript is well-chosen and well-organized. I consider this manuscript suitable for acceptance and publication in the journal. However, before acceptance, it needs a series of revisions as follows:

1- The manuscript needs strong English editing by an English speaker. Some sentences and parts cannot be understood by the readers in the current form. Therefore, it is recommended that the manuscript be carefully edited by an English-speaking speaker.

2- In the whole manuscript, it is better for the authors to use the term REE. Please use not from term "REEs" in the manuscript.

3- For geochemical parameters, use "/" instead of ":". For example, use LREE/HREE instead of LREE:HREE.

4- Use the word "aims" instead of the word "objectives" in line 130.

5- Line 134: Note that Y and Sc are included in the HREE category. Be sure to correct this sentence.

6- I think it is better to use ppm instead of mg kg-1 throughout the manuscript.

7- Please show the correlation coefficients between the elements to two decimal places. For example, r = 0.66.

8- In the introduction section, there are only details about rare earth elements and the names of the elements and their use in industry. It is better to define bauxite in one paragraph in the introduction section and then refer to the studies conducted on the characteristics of rare earth elements in the karst bauxite deposits. Finally, the results of these studies should also be mentioned. This can make the date written from local to international and become very attractive to readers. Below is a paragraph that can improve the manuscript:

The term bauxite was first used by Berthier in 1821 referring to alumina-rich sediments in the Les Baux region of France (Bárdossy, 1982). Al (oxyhydr)oxides are the dominant mineral components of bauxite ore (Bárdossy, 1982), together with variable amounts of Fe and Ti (oxyhydr)oxides. Bauxite deposits are classified into two main categories: (1) bauxite deposits on aluminosilicate bedrocks, and (2) those on carbonate bedrocks, known as karst-type bauxite deposits (Bárdossy, 1982; Bárdossy and Aleva, 1990).

In recent years, there have been studies of the distribution and fractionation of rare earth elements (REE), and factors controlling their distribution and mobility in bauxite deposits worldwide. These studies show that the distribution, fractionation, and enrichment of REE in karst bauxite deposits is controlled by a number of factors, including the chemical composition of the protolith and bedrocks (Mondillo et al., 2022), physicochemical conditions in the weathering environment (Eh–pH) (Reinhardt et al., 2018; Yang et al., 2019; Abedini and Khosravi, 2024), fluctuations of the water table (Radusinovi´c et al., 2017), the presence of organic and inorganic ligands in the soil (Liu et al., 2020; Abedini and Khosravi, 2023), bacterial activity, mineral species in bedrocks (Yuste et al., 2015; Gamaletsos et al., 2019), adsorption, scavenging (Hanilci, 2013; Khosravi et al., 2017), climatic conditions (Mongelli, 1997; Abedini et al., 2024), and chemical properties of elements during weathering (liu et al., 2016).

9- The Conclusions section needs to be improved. Please add more information to this section of the manuscript.

10- In total, the number of references used in this manuscript is small. Please add more up-to-date references to the manuscript, especially in the geochemical interpretations.

Reviewer #2: the manuscript needs corections and modifications. Added in the body of the manuscripts comments and suggestions to improve the quality to the manuscript to meet the standards of the Journal. Accordingly, the comment must be considered point-by-point.

Reviewer #3: Thank you the authors for an interesting study. I agree with the general message contained in the study - that in my experience, portable and in-field instrumentation will likely be the future of geochemical (and similar) exploration, because of precisely the limitations the authors have highlighted in the study.

I have only some specific questions or comments:

(1) Lines 204 to 205 - why did the authors choose ALR and specifically using Si as the denominator. I think this should be made much more explicit, as this choice affects modelling outcome later, because it is a specific choice in data transformation. Please clearly explain the rationale.

(2) Lines 213 to 214 - I understand that the authors assumed 2-sigma values from their pXRF instrument. This reads poorly and sounds like it lacked rigor. Please find the instrument manufacturer and obtain the actual instrumental precision, then report it without speculating.

(3) Lines 219 to 221 - why specifically those hyperparameters? Usually in machine learning, these hyperparameters are explored over ranges using cross-validation (and a train-test split), and the best model is then selected using the grid-searched results (e.g., gridsearchcv function in scikit-learn in Python). Following this section, the authors indicated a 15% validation dataset - was this part of the original dataset and if so, was the original grid search conducted using 85% of the data? Please clarify this. You also mention binned stratification - please clarify on what was binned (e.g., concentrations) and how. These documentations will provide the readers to replicate your method and workflow as needed.

(4) I found the reporting of primarily MAE a bit odd, as it is very good for tuning models (although MSE might be as good or better depending on the context), it is not very good for communicating the quality of models to readers. In figure 8, the authors show the R-squared values for various predictions - please also show the R-squared values in the performance data (table 1) in summary of the model's performance.

(5) Lines 273 to end of paragraph - the authors mentioned multidimensional scaling and principal component analysis were used in comparison to UMAP. I would suggest the authors show these results in a figure, as I cannot imagine what the differences look like by the text alone. Also, the authors should specify the type of multidimensional scaling used - was it metric or non-metric, and if so, what metric was used. Using a non-metric version of multidimensional scaling, only the relative ordering will be preserved - so I am not surprised if the authors used this form, then the separation would not be sufficient for geochemistry in general. If the metric form was used, then using the Euclidean distance metric would work the best with ALR-transformed geochemical data. I would be curious to see the results in this case.

(6) Lines 473 to 474 - can you quantify the deviation in data quality and summarize here? E.g., of the prediction results and/or relative to laboratory or reference field measurements? This is very important scientifically for others wanting to use portable instruments in the field.

(7) Figure 4B - can you plot surface versus subsurface REEs and differentiate by lithology (e.g., differently colorized and markered symbols)? This will help the readers to understand geoscientifically the potential geochemical relationships between surface and subsurface environments.

6. PLOS authors have the option to publish the peer review history of their article (what does this mean?). If published, this will include your full peer review and any attached files.

Reviewer #1: No

Reviewer #2: No

Reviewer #3: No

---

## [Author Response · Author response to Decision Letter 0]

2 Oct 2024

Dear PLosOne, 

Enclosed is the revised manuscript, “Geochemical exploration of rare earth element resources in highland karstic bauxite deposits in the Sierra de Bahoruco, Pedernales Province, Southwestern Dominican Republic”. We appreciate the anonymous reviews provided for this manuscript. The reviewers’ questions at times were challenging, but their feedback instrumental in substantially improving the manuscript. In some cases, reviewers graciously provided additional references and even text to help clarify the underlying context of the manuscript. As a result, the entire manuscript was revised and improved based on the reviewers’ questions and comments. 

Below are responses to reviewers’ comments and questions. Afterward, we included responses to the Editor’s questions. 

Responses to Reviewer #1

1. The manuscript writing was closely reviewed and revised as suggested by the reviewer for greater clarity. 

2. Done. 

3. There seems to be a mixture of presenting the ratio between LREE and HREE with a “:” instead of a “/”. Thanks for pointing that out. Corrections were made throughout the manuscript. 

4. Done. 

5. There is only one list of REE that breaks them up into light and heavy categories, existing on L. 314-315 of the original manuscript. In that list, Y and Sc are both listed as light REE. L. 134 represents the beginning of the field measurement section. 

6. We agree that representing REE concentration can be useful when expressed generally. However, the use of mg kg-1 in this manuscript represents an explicit unit of measurement by which the REE concentrations were determined. Specifically, it is unambiguous regarding how the REE mass was normalized. In other REE exploration studies, measurements of both sediments and waters exist, where unambiguous representation of REE concentration is also required. Stating our concentrations in units of mg kg-1 allows us to avoid any ambiguity in REE representation, and demonstrates that additional sampling “modes “are available for conducting the exploration. 

7. Done

8. We appreciate the helpful introductory paragraph on bauxites provided by the reviewer and have incorporated it into the text.

9. The Conclusions section was expanded with a more extensive synopsis of the study. 

Responses to Reviewer #2 

L. 15. The abstract possesses the PLOS recommended 300-word limit. 

L. 21. Respectfully, we disagree. This hypothesis refers to one of the more important themes emphasized throughout the manuscript. This investigation is not only researching the viability of pXRF for capturing the full range of REE measurements in the field (essentially improving upon the intrinsic capabilities of the instrument, especially overcoming limitations associated with quantifying the total of all 15 REEs), but the research is also focused on utilizing the geochemical information to geolocate the direction and orientation where we expect the REE concentrations to be the most elevated. Thus, as the title of the manuscript implies, REE distributions track the weathering intensity-driven developmental or morphological cues (accessible via hand digging given the remoteness of the area) of the soils, both in terms of subsurface development and elevation-driven climatic shifts observed throughout the Sierra de Bahoruco Mountains. To be clear, we have yet to find any papers that view REE distributions concerning soil morphological developmental features, aside from acknowledging the different regional soil “types” that may be involved. Also, note that the abstract was revised to no longer contain the hypothesis statement.

L. 21. REE was used consistently throughout the manuscript. 

L. 26. To reduce the number of words in the Abstract as recommended by the reviewer, Fig. 2A has been revised to show the lat-lon coordinates of the site. Also, the coordinates were added to a new table in the revised manuscript (see below). 

L. 41. Fixed the redundant definition of the REE abbreviation. 

L. 132. We are unclear as to why the reviewer considers the Introduction section “too long”. Assuming 2.5 to 3 manuscript pages per printed page of PLoS One, we estimate that the Introduction section in the manuscript would occupy < 1.5 printed pages in the printed article. If there is information that the reviewer views as irrelevant or unimportant, the reviewer did not point out any particular section of text. Nonetheless, we attempted to make the Introduction section more concise. Note that this comment contradicts another reviewer’s comments to add a more detailed geological section. We did our best to reconcile the different requests. 

L. 134. Following the reviewer’s recommendation, a table was added (now Table 1) that lists samples by their bauxite deposit group and their corresponding geospatial coordinates, elevation, depth of surface and subsurface samples, texture and color. This change was important to stay within the 300-word limit of the manuscript’s Abstract section. 

L. 136. Done. See comment for L. 26. 

L. 140. Text describing the observed soil profiles was included in the (i) revised caption for Fig. 2B and (ii) at the beginning of the Results section. The exact depths of the subsurface B horizons were not recorded given the difficulty of permeating the dense clay layer. Thus, we assume that the B horizons began at the limit of sampling. 

L. 153. Citation added. Thanks!

L. 157. We weren’t sure how to interpret the reviewer’s comments here. I believe the reviewer is commenting that a color change in the profile does not represent a method for indicating increasing clay content – but not completely sure. Attempting to interpret the reviewer’s comments, our response is that this text does not refer to a method but in fact, is an observation consistently reported at these sites. 

L. 161. Changed. 

L. 163. All references in the revised text were reformatted to the PLoS format. Thanks! 

L. 168. Citation added. Thanks, again! 

L. 185. Changed. 

L. 186. Added text noting that the proprietary standards refer to analytical standards included with the probe and meter. 

L. 188. Company reference added. Thanks!

L. 197. It is difficult to discern the reviewer’s comment here. The previous sentence identified that the soil characterization data was transformed using CoDA to correct for geometric distortions in the data – a justification included in the text and explained with literature citations. The text starting on L. 197 describes the overall workflow associated with these geometric transformations, including data editing, zero imputation, and “closing” the compositions. These transformations represent precursor steps to the ML modeling. We added an introductory sentence that hopefully helps clarify this section further. 

L. 218. Again, it is unclear what the reviewer is communicating here. The first sentence in the paragraph introduces UMAP as an effort to study the clustering of samples based on the geochemical information. There is no “scientific statement” per se, only a description of the method and workflow, which results are explained later in the manuscript. It was our hope that further partitioning out the modeling section with additional headers will help to alleviate the reviewer’s confusion. 

L. 251. It is difficult to reconcile the reviewer’s complaint about the length of the Results section with the numerous requests for more detail. Thus, we endeavored to me more concise in our explanations. 

L. 292. The relevant figure was referred to at the end of the sentence – Fig. 4A. We added “soil characterization” to be more specific about which data to be discussed in this section. 

L. 392. To be clear, we intentionally separated the Results and Discussion sections instead of combining them into a single section. In our opinion, such a division makes it easier to discuss the data directly in the Results section, carefully going through each figure and table, comparing directly, point-for-point with the available scientific literature. Under this format, the Results section will be less speculative or more explicit to the immediate data at hand, which in this manuscript, occupied the previous 5 pages of text. Consequently, the Discussion section is more general, discussing information across the entire manuscript, containing broader interpretations and implicit representations of the overall message contained in the data. Thus, it seems that what the reviewers is interpreting as a non-specific Results section is actually the intended Discussion section, as outlined above. 

L. 484. Done. 

L. 495. Based on the PLoS One website, https://journals.plos.org/complexsystems/s/submission-guidelines#loc-author-contributions, “Those who contributed to the work but do not meet our authorship criteria should be listed in the Acknowledgments with a description of the contribution”. This section was edited to emphasize the contribution of those individuals listed. The remaining text represents a public disclaimer, and acknowledgment of the enterprise research and development programs in which this project was executed and is required by the U.S. government and ERDC to be included in all published work. 

L. 511. Author contributions were expanded based on the reviewer's suggestion. Thanks. 

Fig. 3A. This figure was not discussed in the Discussion section but in the Results section, starting on L. 255 in the previous version of the manuscript. We added some clarification regarding utilizing the breakout of the deposit groups in the ternary plots based on the bauxite definitions provided by Bardossy. 

Fig. 4A and B. This figure represents a standard boxplot. A general description of the programming language and visualizations was provided on L. 176 of the revised manuscript. 

Fig. 7 A-C is part of a multi-paneled plot. It is possible to combine plots 7A-C, but doing so may compromise the details of each individual plot, such as making the single plot so cumbersome that the diagonal, prediction line, corresponding confidence intervals, and equation are obscured. 

Responses to Reviewer #3. 

1. L. 204-205. Explanation was provided for our decision to conduct the transforms. Here, we clarified that the centered log ration (clr) transform was used for the bulk composition data while the additive log ratio (alr) was used for the pXRF data. The reasoning here is that the acid digestions used for generating the bulk composition don’t quantitatively consume Si. Thus, being unreliable, that column was removed before transforming the data. However, the pXRF does measure Si, so we normalized the data by Si as it is a common practice to do so for alumnosilicates. The reviewer is correct that the choice of transformation does influence the data; however, it was more important for us to remove the geometric distortion and get the correction. In our view, the choice of transformation method wasn’t all that important since the pXRF model was calibrated based on the total REE measured in the acid digest data, so it wasn’t necessarily important that the data columns line up between the bulk composition (which excluded the REE columns) and the pXRF data. 

2. L. 219-221. More explanation was provided. We chose the approach for the zero interpolations as it seemed most reasonable based on the way the data is presented on the pXRF spectrometer. Ultimately, the choice of lower detection limits don’t matter all that much in the zero-interpolations, as that data contributes nominally to the overall signature. This is especially the case of the variable is largely comprised of NDs as it will most likely fall under the 5% our very conservative 5% missing data threshold, and will likely be removed before completing the zero interpolation anyway. But, we appreciate the question. 

3. L. 2019-220. See the new Table 2 where the hyperparameters and initial estimates are listed. Also, see the revisions for a newly labeled section, called “ML regression modeling” as well as a new description of an optimized resampling scheme to show the potential best estimate of the model for the current dataset. 

4. We replaced the MAE parameter with RMSE, with these values and the optimized R-squared parameters included in a new Table. 3. 

5. A new Supporting Information document was created for this manuscript, which includes description and the plots from PCA and MDS analyses. These analyses were performed early in this work using the pXRF. 

6. It seems that we did not exactly quantify this difference – it became obvious to us early in the work as the fewer REE were detected when the sample was anlayzed in the field vs. after processing in the lab. The outcome was sensible given our years of published Synchrotron-based x-ray work, which requires rigorous sample preparation. Additionally, those REE that were detected in the field were quantified at much lower concentrations when analyzed on the processed samples. Actually, we previously cited a publication in the original manuscript but for the revised manuscript, provided some additional explanation. 

7. Unfortunately, we cannot as that data doesn’t exist. This was very much a proof of concept study, conducted on a shoe-string budget. However, thank you for the suggestion as we are commencing a much larger study beginning October 2024, and we can include lithological descriptors as a label to the new datasets. 

Responses to the Editor

1. Based on the combined comments from the Editor and the three reviewers, the Introduction section was extensively revised. Some reviewers claimed the section was too long, others claimed that the Introduction required additional detail, such as requesting an added section detailing the areas geology, and a section written by one of the reviewers on bauxite. We did our best to reconcile these conflicting recommendations, attempting to make the Introduction section more concise, yet better focused on the specific information of interest to the reviewers. 

2. See text added to the Introduction beginning on L. 50 of the revised manuscript which describes the geological setting of the study area described in this manuscript. 

3. Done – see above response to the reviewer’s comment on L. 21. 

4. Done. Extensive additions have been made to the manuscript casting greater detail on the ML modeling aspects. Also, see Table 3 in the revised manuscript. 

5. Done, please see the expanded Discussion section. 

6. Done, please see the expanded Conclusions section. 

7. The manuscript has been extensively revised to better meet these requirements.

---

## [Decision Letter · Decision Letter 1]

28 Oct 2024

PONE-D-23-43088R1Geochemical exploration of rare earth element resources in highland karstic bauxite deposits in the Sierra de Bahoruco, Pedernales Province, Southwestern Dominican RepublicPLOS ONE

Dear Dr. Chappell,

Thank you for submitting your manuscript to PLOS ONE.** After careful review, we are pleased to inform you that your manuscript can be accepted for publication, pending minor revisions. To enhance readability and further clarify your findings, we recommend please consider dividing the abstract into sub-sections, such as "Introduction," "Objectives," "Materials and Methods," "Key Results," and "Concluding Remarks." This format will improve clarity and allow readers to quickly grasp the significance of your work.**

**We believe these minor changes will contribute to the overall quality of your manuscript. Once completed, please submit your revised manuscript at your earliest convenience.**

**Thank you again for your submission. We look forward to seeing the final version and are excited to share your work with the broader research community.**

We look forward to receiving your revised manuscript.

Kind regards,

Rizwan Sarwar Awan

Academic Editor

PLOS ONE

**Journal Requirements:**

Reviewers' comments:

Reviewer's Responses to Questions

**Comments to the Author**

1. If the authors have adequately addressed your comments raised in a previous round of review and you feel that this manuscript is now acceptable for publication, you may indicate that here to bypass the “Comments to the Author” section, enter your conflict of interest statement in the “Confidential to Editor” section, and submit your "Accept" recommendation.

Reviewer #1: All comments have been addressed

Reviewer #2: All comments have been addressed

2. Is the manuscript technically sound, and do the data support the conclusions?

Reviewer #1: Yes

Reviewer #2: Yes

3. Has the statistical analysis been performed appropriately and rigorously? 

Reviewer #1: Yes

Reviewer #2: Yes

4. Have the authors made all data underlying the findings in their manuscript fully available?

Reviewer #1: Yes

Reviewer #2: Yes

5. Is the manuscript presented in an intelligible fashion and written in standard English?

Reviewer #1: Yes

Reviewer #2: Yes

6. Review Comments to the Author

**Reviewer #1: **I carefully reviewed the responses to the comments. The manuscript is well improved and can be accepted for publication in the journal.

**Reviewer #2: **The manuscript is improved but it can be better if the abstract can be sub sectioned to introduction, objectives, materials and methods a

Nd interesting results beside a concluding remark. Further more, the conclusion section can include a research problem in the conclusion section for the future researchers.

7. PLOS authors have the option to publish the peer review history of their article (what does this mean?). If published, this will include your full peer review and any attached files.

Reviewer #1: No

Reviewer #2: **Yes: **Professor Dr Yasser El-Nahhal

---

## [Author Response · Author response to Decision Letter 1]

7 Nov 2024

The changes made in this manuscript were in response to the comments made by the Editor and Reviewer 2, related to create subheadings for the Abstract section. In addition, Reviewer 2 requested that the Conclusions section of the manuscript include a research problem for future researchers. 

Throughout this revision, we made minor changes to the manuscript as an additional careful review showed some inconsistencies in our explanations (held over from previous revisions), such as older descriptions of the models before changing to representing the error by RMSE values, and the inclusion of the resampling exercise. In particular, improvements in the ML modeling made the pXRF and color sensor comparable in terms of their prediction accuracies – a significant result now included in the Discussion section given the 1000x cost difference between the two sensors. In addition, we beefed up some of the discussion based on some of the newer literature about karstic bauxites in the area as well as potential insights from the scientific literature regarding Fe-oxyhyroxy – REE speciation that may influence their efficient extraction and recovery from the bauxites. 

From these revisions, a higher-priority research question (the question itself requested by Reviewer2) emerged as the need for an efficient way to predict REE speciation (in addition to concentration, the other important component driving the economic feasibility of a resource) based on the output from the portable sensors. Extensive and geochemical speciation is applied typically to data developed under controlled laboratory conditions, but rarely if ever applied “retroactively” to the portable sensor responses. This, in part may be due to the general sense of the inherent error associated with these portable sensors, which we largely overcame computationally. For many heavy metal “contaminants”, the literature shows that soil speciation is often driven by the combination of concentration and the chemistry of a particular soil “type”. With REE concentrations exceeding 0.1 % in many cases within this deposit, the distribution of REE species may indeed be diverse and concentration-dependent.

---

## [Editor Report · Decision Letter 2]

12 Nov 2024

PONE-D-23-43088R2Geochemical exploration of rare earth element resources in highland karstic bauxite deposits in the Sierra de Bahoruco, Pedernales Province, Southwestern Dominican RepublicPLOS ONE

Dear Dr. Chappell,

Thank you for submitting your manuscript to PLOS ONE, after careful review, your manuscript titled "Geochemical exploration of rare earth element resources in highland karstic bauxite deposits in the Sierra de Bahoruco, Pedernales Province, Southwestern Dominican Republic" has been accepted for publication in PLOS ONE, pending minor revisions.

The only revision required is to remove the sub-heading of the abstract, without altering its content. We believe that this small adjustment will enhance the presentation of your manuscript.

Once the revision has been made, kindly submit the updated version of your manuscript at your earliest convenience. Thank you again for submitting your research to PLOS ONE. We look forward to receiving your revised manuscript and sharing your valuable findings with the broader scientific community.

We look forward to receiving your revised manuscript.

Kind regards,

Rizwan Sarwar Awan

Academic Editor

PLOS ONE
---

## [Author Response · Author response to Decision Letter 2]

18 Nov 2024

The changes made in this manuscript were in response to the comments made by the Publisher to go back and remove the Abstract subheadings previously recommended by the Editor and Reviewer 2.

---

## [Editor Report · Decision Letter 3]

21 Nov 2024

Geochemical exploration of rare earth element resources in highland karstic bauxite deposits in the Sierra de Bahoruco, Pedernales Province, Southwestern Dominican Republic

PONE-D-23-43088R3

Dear Dr. Chappel,

We’re pleased to inform you that your manuscript has been judged scientifically suitable for publication and will be formally accepted for publication once it meets all outstanding technical requirements.

Kind regards,

Rizwan Sarwar Awan

Academic Editor

PLOS ONE
---

## [Editor Report · Acceptance letter]

21 Dec 2024

PONE-D-23-43088R3 

PLOS ONE

Dear Dr. Chappell, 

I'm pleased to inform you that your manuscript has been deemed suitable for publication in PLOS ONE. Congratulations! Your manuscript is now being handed over to our production team.

Kind regards, 

on behalf of

Dr. Rizwan Sarwar Awan 

Academic Editor

PLOS ONE